# On Designing General and Expressive Quantum Graph Neural Networks with Applications to MILP Instance Representation

**Xinyu Ye[1], Hao Xiong[1], Jianhao Huang[1], Ziang Chen[2], Jia Wang[1], Junchi Yan[1]***

1. Shanghai Jiao Tong University, Shanghai, China.
2. Massachusetts Institute of Technology, Massachusetts, United States.
{xinyu_ye,taxuexh,huang_jh2021}@sjtu.edu.cn
ziang@mit.edu,{jiawang,yanjunchi}@sjtu.edu.cn

## Abstract

Graph-structured data is ubiquitous, and graph learning models have recently been extended to address complex problems like mixed-integer linear programming (MILP). However, studies have shown that the vanilla message-passing based graph neural networks (GNNs) suffer inherent limitations in learning MILP instance representation, i.e., GNNs may map two different MILP instance graphs to the same representation. In this paper, we introduce an expressive quantum graph learning approach, leveraging quantum circuits to recognize patterns that are difficult for classical methods to learn. Specifically, the proposed General Quantum Graph Learning Architecture (GQGLA) is composed of a node feature layer, a graph message interaction layer, and an optional auxiliary layer. Its generality is reflected in effectively encoding features of nodes and edges while ensuring node permutation equivariance and flexibly creating different circuit structures for various expressive requirements and downstream tasks. GQGLA is well suited for learning complex graph tasks like MILP representation. Experimental results highlight the effectiveness of GQGLA in capturing and learning representations for MILPs. In comparison to traditional GNNs, GQGLA exhibits superior discriminative capabilities and demonstrates enhanced generalization across various problem instances, making it a promising solution for complex graph tasks.

## 1 Introduction

Mixed-integer linear programming (MILP) serves as a general optimization formulation applicable to diverse real-world optimization scenarios, such as transportation (Schouwenaars et al., 2001), scheduling (Floudas & Lin, 2005b), and production planning (Askari-Nasab et al., 2011).

MILP aims to minimize a linear objective function while adhering to linear constraints. Apart from the classic non-learning solvers that often resort to heuristics, recent learning-based models have been actively studied, and graph neural networks (GNNs) are considered a suitable backbone to represent the mappings for MILP instances in various stages of MILP solving processes (Khalil et al., 2022; Gupta et al., 2022; Wang et al., 2023; Li et al., 2025). A MILP instance can be regarded as a weighted bipartite graph with node features, as illustrated in Fig. 1 and Fig. 2. Due to the beneficial property of GNN, permutations on variables or constraints of a MILP do not essentially change

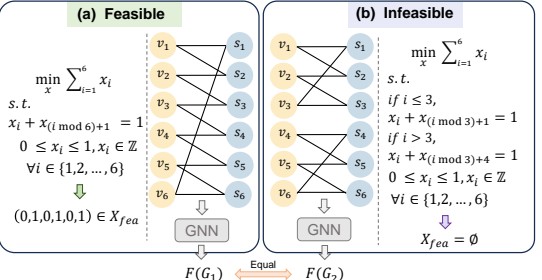

Figure 1: A pair of *GNN-intractable* MILPs $G_1$ and $G_2$. **(a)** is feasible and **(b)** is infeasible. Although their edge connectivity and feasibility are different, GNNs embed them into the same representation.

*Correspondence author who is also affiliated with Shanghai Innovation Institute. Work was partly supported by NSFC (62222607), Shanghai Municipal Science and Technology Major Project (2021SHZDZX0102).

Table 1: Comparison of QNNs for *classical* graph data on several aspects: whether the models provide implementable circuits, enjoy permutation equivariance, consider multi-dimensional node or edge features, utilize quantum (Q) layers or classical (C) layers, and the manner of readout. Compared to other methods, our proposed model addresses multiple aspects, can be applied to both node-level and graph-level tasks, and achieves advantage validation over GNNs in learning graph representations.

| Method | Quantum Circuit Embodied | Permutation Equivariance | Attribute | Layer | Readout | Application |
|---|---|---|---|---|---|---|
| QGNN (Verdon et al., 2019) | ✗ | ✓ | — | Q | Tomography | Learning Hamiltonian Dynamics & Graph Isomorphism Classification |
| QGCN (Zheng et al., 2021) | ✓ | ✗ | Node & Edge | Q | Estimation | Image Classification |
| egoQGNN (Ai et al., 2022) | ✓ | ✗ | Node | Q & C | Tomography | Graph Classification |
| EQGC (Mernyei et al., 2022) | ✗ | ✓ | Node | Q & C | Estimation | Synthetic Cycle Graph Classification |
| **Ours** | ✓ | ✓ | Node & Edge | Q | Estimation | Graph Classification & Regression |

the problem itself. This can prevent the model from overfitting the variable/constraint orders in the training data. However, a recent study (Chen et al., 2023) revealed that the classical GNNs based on the message-passing mechanism suffer fundamental limitations in graph representation, especially for MILP graphs, *i.e.*, GNNs do not have sufficient power to distinguish some different instances. Specifically, as shown in Fig. 1, two different MILP instances can be eventually embedded into the identical representation by GNNs, thus failing to predict the feasibility of MILP. Moreover, in real-world scenarios, there are numerous MILPs that GNNs cannot distiguish (Chen et al., 2023), which means that practitioners using GNNs may not benefit from this.

As we can see, classical GNNs face fundamental limitations in learning graph representation. To overcome this challenge, we turn to quantum machine learning (QML), which is an emerging field that combines the power of quantum computing with the capabilities of machine learning. QML has shown immense potential in recent years, such as recognizing patterns that are intractable using classical methods (Biamonte et al., 2017). In this paper, we aim to associate nodes with qubits and edges with quantum entanglement, investigating graph structures that are indistinguishable from classical GNNs and exploring effective approaches for constructing quantum graph neural networks.

Designing a quantum learning framework for unstructured data, such as graph data, is still in its nascent stages. Although some quantum neural networks (Verdon et al., 2019; Zheng et al., 2021; Ai et al., 2022; Mernyei et al., 2022; Yan et al., 2023) designed to handle graph data have been proposed, they struggle to effectively apply to complex graph tasks such as MILP graphs because they fail to address the following challenges simultaneously. **i)** Since complex graph data usually contains node and edge features, it is important to design a reasonable node/edge feature encoding strategy that enables the quantum circuit to utilize these features to learn effective node representations.**ii)** As mentioned earlier, classic GNNs possess the beneficial property of *permutation equivariance*. However, designing a quantum circuit that conforms to permutation equivariance is nontrivial and requires careful design. **iii)** Unlike building hybrid classical-quantum layers, designing full quantum circuits for classical graph data and verifying them in practical graph tasks is challenging. Moreover, we provide a detailed discussion of related works in Appendix A and Table 1 here as a summary.

To tackle the above challenges, we propose a **general quantum graph learning architecture named GQGLA**. This method aims to provide a quantum solution for complex graph tasks like learning MILP representation and to demonstrate superior discriminative power over classical GNNs. The generality of GQGLA is reflected in the following five aspects. **i)** GQGLA can encode node and edge features. We design learnable features associated with each feature and flexible encoding schemes to better learn node representations. **ii)** With the proposed parameter-sharing strategy, all the layers in GQGLA possess the beneficial property of permutation equivariance, and we provide the theoretical proof in Sec. 4.3. **iii)** GQGLA can incorporate an optional auxiliary layer to enhance the expressive power of the model. **iv)** The types of quantum gates in GQGLA are adjustable under our configuration principle, allowing flexible use across different scenarios. **v)** By designing different measurement layers, GQGLA can be applied to graph tasks at various levels, such as graph classification, graph regression, and node property prediction. **We summarize the main contributions:**

**1)** To explore the potential of quantum machine learning for learning graph-structured data, we present a general quantum graph learning architecture (GQGLA). The method is based on fully quantum circuits compatible with current devices and is capable of effectively encoding the features of nodes and edges while ensuring node permutation equivariance. GQGLA can flexibly create different circuit

structures to meet various expressive requirements and downstream tasks. Moreover, we theoretically prove the permutation equivariance of GQGLA.

**2)** We implement GQGLA in an application for learning MILP representation, which is evaluated on three tasks: predicting feasibility, the optimal value, and the optimal solution. Numerical experiments demonstrate the advantage of GQGLA over classic GNNs in learning MILP representations. GQGLA has better discriminative power to handle GNN-intractable MILPs. It also shows that GQGLA has better generalization and uses fewer parameters.

**3)** The superior discriminative power of GQGLA over GNNs not only provides a promising solution for more challenging graph tasks but also shows the practicability of current quantum machine learning methods. This will encourage further exploration of the potential of QNNs over their classical counterparts.

## 2 PRELIMINARIES

We provide the basics of quantum computing and quantum machine learning in a simple and understandable way in Appendix B, ensuring that readers with a background in linear algebra but not familiar with quantum computing can gain a basic understanding of the quantum technologies used in our paper. The matrix forms of all quantum gates mentioned in the paper are provided in Table 9. Next, we will introduce the specific form of MILP graphs and the types of MILP datasets.

### 2.1 MILP GRAPHS AND THE LIMITATION OF CLASSIC GNNs

**Graph Representation for MILPs.** An instance is defined as follows $A \in \mathbb{R}^{p \times q}$, $b \in \mathbb{R}^p$, $c \in \mathbb{R}^q$:

$$\min_{x \in \mathbb{R}^q} \ c^\top x, \quad \text{s.t. } Ax \circ b, \ l \leq x \leq u, \ x_i \in \mathbb{Z}, \ \forall \, i \in I, \tag{1}$$

where $l$ and $u$ represent the upper and lower bounds on variables, where $l \in (\mathbb{R} \cup \{-\infty\})^q$, $u \in (\mathbb{R} \cup \{+\infty\})^q$ and $\circ \in \{\leq, =, \geq\}^p$. Consistent with Chen et al. (2023), the mathematical operators $\{\leq, =, \geq\}$ are mapped into numerical value $\{0, 1, 2\}$, respectively. $I \subseteq \{1, \cdots, q\}$ represents the index set of integer variables. The *feasible solution* is defined as the set $X_{fea} = \{x \in \mathbb{R}^q \mid Ax \circ b, \ l \leq x \leq u, \ x_i \in \mathbb{Z}, \ \forall i \in I\}$, while $X_{fea} = \emptyset$ means the MILP problem is *infeasible*. Feasible MILPs have an *optimal objective value* $y_{obj} = \inf\{c^\top x \mid x \in X_{fea}\}$. If there exists $\hat{x} \in X_{fea}$ such that $c^\top \hat{x} \leq c^\top x, \forall \, x \in X_{fea}$,

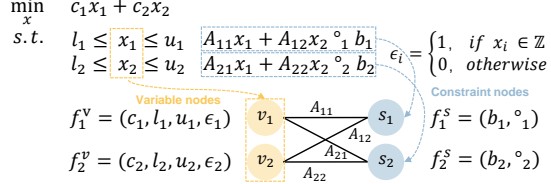

Figure 2: A weighted bipartite graph of a MILP instance. $v_i$ is the variable node associated with feature $f_i^V$ and $s_j$ indicates the constraint node associated with feature $f_j^S$. The edge between $v_i$ and $s_j$ means the $j$-th constraint involves the $i$-th variable.

then $\hat{x}$ is an *optimal solution*. Nevertheless, the optimal solution may not always exist because the optimal objective value can be arbitrarily good, where the MILP problem is *unbounded* and categorized as feasible with an optimal objective value of $-\infty$. Following the protocol in Gasse et al. (2019) and Chen et al. (2023), we formulate MILP as a *weighted bipartite graph* to interpret variable-constraint relationships, as illustrated in Fig. 2. The vertex set is $V \cup S$, where $V = \{v_1, \cdots, v_i, \cdots, v_q\}$ with $v_i$ representing the $i$-th variable and $S = \{s_1, \cdots, s_j, \cdots, s_p\}$ with $s_j$ representing the $j$-th constraint. The edge connected $v_i$ and $s_j$ has weight $A_{i,j}$. Based on Eq. (1), the vertex $v_i \in V$ is associated with a feature vector $f_i^V = (c_i, l_i, u_i, \epsilon_i)$, where $\epsilon_i \in \{0, 1\}$ represents whether variable $v_i$ takes an integer value. The vertex $s_j$ is equipped with a two-dimensional vector $f_j^S = (b_j, \circ_j)$. There is no edge between vertices in the same vertex set ($V$ or $S$). The weighted bipartite graph with node features is named an *MILP-induced graph* or *MILP graph*.

**Classic GNNs may Fail on General MILPs.** Recall that Chen et al. (2023) has shown that GNNs may embed two different (one feasible and one not) MILPs into an identical embedding. In fact, there are infinitely many pairs of MILP instances that can puzzle GNNs. Therefore, Chen et al. (2023) call this class of MILPs that confuse GNNs as *foldable* MILPs, while the rest of the MILPs are named *unfoldable* MILPs. In this paper, we refer to them as *GNN-intractable* MILPs and *GNN-tractable* MILPs, respectively. Fig. 1 gives an example of a pair of MILPs in the GNN-intractable MILP dataset. In this case, $f_i^V = (1, 0, 1, 1)$, for all $v_i \in V$ and $f_j^S = (1, =)$, for all $s_j \in S$. All edge

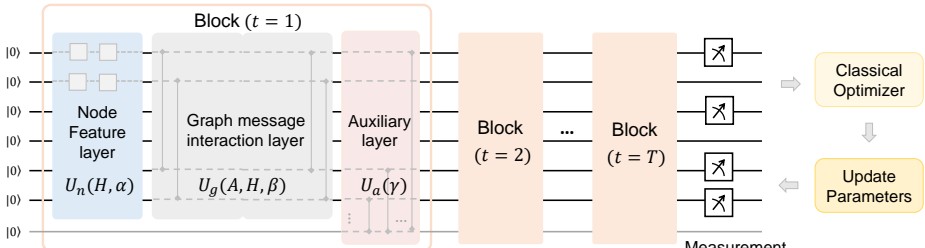

Figure 3: The overall architecture of our GQGLA. The node feature layer encodes and learns node features into the quantum circuit, and the graph message interaction layer contains a variable update layer and a constraint update layer. The auxiliary layer is optional and is used to enhance the model's capacity. All layers are designed to preserve the equivariance of the node permutation.

weights are equal to 1, which means that the only difference between the two bipartite graphs lies in the connectivity of edges. However, these two MILP instances have different feasibility. Fig. 1 (a) is feasible, e.g. $x = (0, 1, 0, 1, 0, 1)$ is a feasible solution, while Fig. 1 (b) is infeasible as there are no integer decision variables that can satisfy the equality constraint $2(x_1 + x_2 + x_3) = 3$. Appendix E.1 illustrates why GNN has no ability to distinguish between them. The limitation of GNNs means that directly applying GNNs to represent general MILPs may fail.

# 3 GENERAL QUANTUM GRAPH LEARNING ARCHITECTURE

Fig. 3 shows the overall framework of GQGLA, which consists of the node feature layer, graph message interaction layer, auxiliary layer, and measurement layer. The first three layers form a block. After the block is iteratively repeated $\Gamma$ times, Pauli-Z measurement is performed. This section introduces the design details of each layer in GQGLA using MILP graphs as an example. The next section presents how GQGLA can be applied to various levels of graph tasks of MILP and demonstrates its permutation equivariance and discriminative power.

## 3.1 NODE FEATURE LAYER

Fig. 4 (a) exemplifies a node feature layer with two variable nodes and two constraint nodes. The variable nodes $v_i$ have four features $(c_i, l_i, u_i, \epsilon_i)$, and the constraint nodes $s_j$ have two features $(b_j, \circ_j)$. Suppose that each qubit encodes two features, as shown in Fig. 4 (c). Specifically, $W(\theta)$ and $D(\theta)$ represent two *different* types of single-qubit quantum gates, including $\{R_X(\theta), R_Y(\theta), R_Z(\theta)\}$, where $\theta$ indicates the parameter of the gate. We can set $\theta$ as either trainable parameters or a constant to encoded. The selection principle of single-qubit quantum gates ($W(\theta)$ and $D(\theta)$) will be described in Sec. 3.4. In this case, the first qubit encodes the features $c_1$ and $l_1$, and the second qubit encodes the features $u_1$ and $\epsilon_1$. That is, the first two qubits are used to

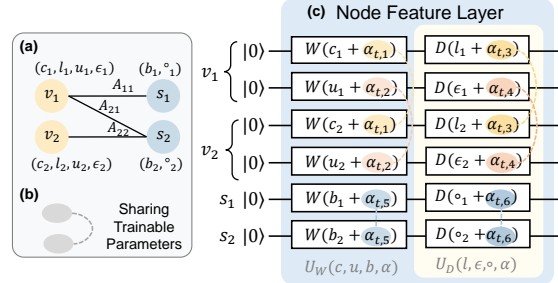

Figure 4: Node features are encoded into the circuit using angle encoding, and each feature is associated with a trainable parameter to learn the node representation. For node permutation equivariance, the same feature dimension (*e.g.*, $c_1$ and $c_2$) shares the identical trainable parameter (*i.e.*, $\alpha_{t,1}$).

represent the node $v_1$. When fewer features are encoded on each qubit, more qubits are used to represent a node, resulting in a richer node representation. However, this also leads to increased model complexity, so we set $\omega$ as a hyperparameter to denote the maximum number of features that each qubit can encode, balancing the speed and accuracy. Fig. 4 shows the case where $\omega = 2$, and we also provide another illustration of $\omega = 4$ in Fig. 8 in Appendix. Moreover, the same feature dimension (*e.g.*, $c_1$ and $c_2$) shares the identical *trainable* parameter (*i.e.*, $\alpha_{t,1}$ in the $W(c_1 + \alpha_{t,1})$ and $W(c_2 + \alpha_{t,1})$). In other words, the trainable parameters are related only to the feature dimension, not to the permutation of the nodes. Thereby, even if the order of nodes changes (*e.g.*, encoding node $v_2$ using the first two qubits), each node's features are assigned the same trainable parameters, ensuring that the learned node feature representation remains invariant. In this way, the node feature layer can preserve the node permutation equivariance, see Sec. 4.3 for proof. The node feature layer encodes

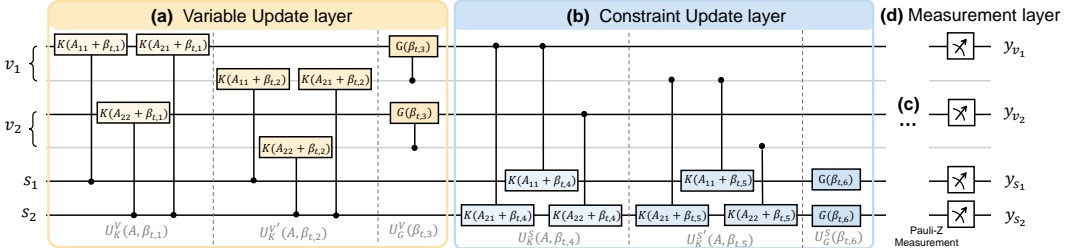

Figure 5: The quantum graph message interaction layer. **(a)** Variable update layer. Controlled-$K$ gates are used to learn the information interaction between variable nodes and constraint nodes, while controlled-$G(\theta)$ gates are utilized to learn the internal interaction between qubits representing the same variable nodes. **(b)** Constraint update layer. Controlled-$K$ gates are turned upside down and are followed by $G(\theta)$ gates to update the representation of constraint nodes. **(c)** Omitted node feature layers and graph message interaction layers. **(d)** Measurement layer using Pauli-Z measurement.

all variable features $H^V \in \mathbb{R}^{q \times 4}$ and constraint features $H^S \in \mathbb{R}^{p \times 2}$ can be defined as:

$$U_x(H, \alpha_t) = U_W(c, u, b, \alpha_t)U_D(l, \epsilon, \circ, \alpha_t). \tag{2}$$

## 3.2 QUANTUM GRAPH MESSAGE INTERACTION LAYER

We employ two-qubit quantum gates to entangle the qubits representing two nodes connected by an edge. By harnessing the mechanism of quantum entanglement, we can learn the information interaction in MILP graphs. Fig. 5 exemplifies the graph message interaction layer for the MILP graph in Fig. 4 (a), which contains three edges. Specifically, this layer includes two parts: the variable update layer and the constraint update layer. We set the two-qubit quantum gates as controlled-$K(\theta) \in \{R_X(\theta), R_Y(\theta), R_Z(\theta)\}$. In the variable update layer, the control qubit of controlled-$K(\theta)$ gate is the constraint node, and its target qubit is the variable node. Different edges share an identical trainable parameter $\beta_t$, as illustrated in the first three gates in Fig. 5 (a). In this case, one variable node is represented by two qubits. Interestingly, the order of controlled-$K$ gates can be arbitrary due to the character of bipartite graphs, *i.e.*, whether the controlled-$K(A_{11} + \beta)$ or the controlled-$K(A_{22} + \beta)$ is applied first does not change the unitary of the $U_K$ layer. We refer to this property as *edge permutation invariance*. In the MILP graph, the edge feature is the edge weight, but GQGLA can also be applied to cases with multi-dimensional edge features, as illustrated in Fig. 9 in the Appendix. Moreover, the controlled-$G$ gate is applied to these two qubits to learn the interaction of *internal* information within one node, where controlled-$G(\theta) \in \{CR_X(\theta), CR_Y(\theta), CR_Z(\theta)\}$. In the constraint update layer, the controlled-$K$ gates are turned upside down and are followed by $G(\theta)$ gates to update the feature representation of constraint nodes. The selection principle of $K$ and $G$ gates will be described in Sec. 3.4. The unitary of the $t$-th graph message interaction layer can be represented by $U_g(A, \beta_t) = U_{gv}(A, \beta_t) \cdot U_{gs}(A, \beta_t)$, with formulas shown in Eq. 10. We also provide a simple and intuitive explanation in Appendix D.2.4. Furthermore, we theoretically prove the node permutation invariance of this layer in Sec. 4.3.

## 3.3 AUXILIARY LAYER

To further enhance the expressiveness of the model, inspired by Wu et al. (2021), we introduce an optional auxiliary layer, which can facilitate the interaction of information within the graph. Specifically, each auxiliary qubit is connected to all other nodes through symmetric two-qubit gates $R_{ZZ}(\gamma)$. In this way, the model can increase the width and number of parameters. For the two qubits representing one variable, trainable parameters $\gamma_{t,1}$ and $\gamma_{t,2}$ are assigned, while the parameter $\gamma_{t,3}$ is assigned to the qubit representing constraints. The unitary of the auxiliary layer is defined as $U_a(\gamma_t)$. We can choose a varying number of auxiliary qubits based on the requirements of different tasks. The node feature layer, graph message interaction layer, and auxiliary layer are regarded as a block. After iterating this block $\Gamma$ times, the unitary matrix of the overall circuit is equal to

$$U_{qgl}(A, H, \Theta) = \prod_{t=1}^{\Gamma} U_x(H, \alpha_t)U_g(A, \beta_t)U_a(\gamma_t), \tag{3}$$

where $\Theta$ contains all trainable parameters $\alpha, \beta,$ and $\gamma$.

### 3.4 CONFIGURATION PRINCIPLE OF GQGLA

In GQGLA, the quantum gates $W(\theta)$, $D(\theta)$, $K(\theta)$, and $G(\theta)$ are selected from $\{R_X(\theta), R_Y(\theta), R_Z(\theta)\} = \{\exp(-\mathbf{i}\theta\sigma_x), \exp(-\mathbf{i}\theta\sigma_y), \exp(-\mathbf{i}\theta\sigma_z)\}$, where $\{\sigma_x, \sigma_y, \sigma_z\}$ are Pauli matrices form a basis for the real vector space of $2 \times 2$ Hermitian matrices. $\{\mathbf{i}\sigma_x, \mathbf{i}\sigma_y, \mathbf{i}\sigma_z\}$ form a basis for the real Lie algebra $\mathfrak{su}(2)$, which exponentiates to the special unitary group $SU(2)$. When two quantum gates are adjacent in GQGLA, they are expected to be of different types. This enables the encoding of various features or trainable parameters on different bases, thereby enriching the information encoded in the circuit and preventing the quantum circuit from confusing different features. There are six possible configurations of GQGLA, which are summarized in Table 10.

## 4 GQGLA WITH APPLICATION TO MILP

As the application of GQGLA, predicting feasibility, optimal objective value, and optimal solution of MILP graphs can be regarded as the task of graph classification, graph regression, and node property prediction, respectively. As we can see, learning the MILP representation is a sufficiently complex task that fully utilizes GQGLA, thereby comprehensively showcasing its capabilities.

### 4.1 MEASUREMENT LAYER

As shown in Fig. 5 (d), we add a measurement layer at the end of the quantum circuit. After the measurement operation, the quantum information can be translated into classical information. In the measurement layer, we measure only the first qubit representing each node using Pauli-Z measurement. That is, for $q$ variable nodes and $p$ constraint nodes, GQGLA outputs $q + p$ results. The output of the GQGLA is defined as $\Phi(A, H, \Theta) = \{\langle 0|U_{qgl}^{\dagger}(A, H, \Theta)O_i U_{qgl}(A, H, \Theta)|0\rangle\}_{i=1}^{q+p}$, where $O_i$ represents $i$-th observable. For predicting the feasibility and optimal value of MILP graphs, we define $\phi_{\text{fea}}(A, H) = \sum_{i=1}^{q+p} \Phi(A, H, \Theta_{fea})_i$, $\phi_{\text{obj}}(A, H) = \sum_{i=1}^{q+p} \Phi(A, H, \Theta_{obj})_i$. For predicting the optimal solution, we define $\phi_{\text{sol}}(A, H) = \{\Phi(A, H, \Theta_{sol})_i\}_{i=1}^{q}$. As can be seen, the three tasks use the same circuit structure of GQGLA but employ different trainable parameters and different ways to utilize the information obtained by measurements.

### 4.2 TRAINING AND TESTING

**For predicting the feasibility,** $\hat{y}_{fea} = \phi_{\text{fea}}(A, H)$, we utilize the negative log-likelihood as the loss to train GQGLA. In the testing, we set an indicator function $\mathbb{I}_{\hat{y}_{fea}>1/2}$ :

$$\mathbb{I}_{\hat{y}_{fea}>1/2} = \begin{cases} 0, & \hat{y}_{fea} \leq 1/2 \\ 1, & \hat{y}_{fea} > 1/2 \end{cases} \tag{4}$$

to calculate the *error rate*: $\frac{1}{M}(\sum_{m=1}^{M} y_{fea}^m \cdot \mathbb{I}_{\hat{y}_{fea}>1/2}^m)$, where $M$ indicates the number of tested MILP instances, to evaluate the number of correct predictions for feasibility.

**For predicting the optimal solutions,** $\hat{y}_{sol} = \lambda\phi_{\text{sol}}(A, H)$, where $\lambda$ is the maximum range of variables of the training sample, *i.e.*, $\max\{\{\{abs(l_i^e), abs(u_i^e)\}_{i=1}^{q}\}_{e=1}^{E}\}$. We use the *mean square error* as the training and testing metric: $\frac{1}{Mq}\sum_{m=0}^{M-1} \|y_{sol} - \hat{y}_{sol}^m\|_2^2$, where $y_{sol}^m$ is the ground truth.

**For predicting the optimal values,** $\hat{y}_{obj} = \delta\lambda\phi_{\text{obj}}(A, H)$, where $\delta = \max\{\{\{c_i^e\}_{i=1}^{q}\}_{e=1}^{E}\}$ is the maximum range of coefficients of training sample. We also use the *mean square error* to train or evaluate, *i.e.*, $\frac{1}{M}\sum_{m=0}^{M-1}(y_{obj}^m - \hat{y}_{obj}^m)^2$.

### 4.3 PERMUTATION EQUIVARIANCE AND INVARIANCE

Permutation equivariance and invariance are fundamental requirements for graph neural networks. In this section, we will demonstrate that the proposed quantum graph learning architecture possesses these desirable properties. Let $G = (A, H^{\text{V}}, H^{\text{S}})$ represent a MILP graph with $q$ variable nodes and $p$ constraint nodes. $\pi_q$ and $\pi_p$ are the permutations of variable nodes and constraint nodes, respectively. $\Lambda_q \in \mathbb{B}^{q \times q}$ and $\Lambda_p \in \mathbb{B}^{p \times p}$ are permutation matrices representing a permutation $\pi_q$ and $\pi_p$, respectively. Additionally, we define a permutation $\pi_{q+p}$ on the $(q + p)$ elements to indicate the overall permutation on all nodes of the MILP graph. $\Lambda_{q+p}$ is the permutation matrix of the

permutation $\pi_{q+p}$. $\widetilde{\Lambda}$ is the *unitary* representation of $\pi_{q+p}$, which indicates the permutation of qubits representing nodes. Based on this, we give the definition of a permutation equivariant circuit.

---

**Defintion 1.** *(Permutation Equivariant Circuit).* *A parameterized quantum circuit $U$ is permutation equivariant for MILP graphs $(A, H^V, H^S)$ if it satisfies*

$$U(\Lambda_p A \Lambda_q, \Lambda_q H^V, \Lambda_p H^S, \Theta) = \widetilde{\Lambda}^\dagger \cdot U(A, H^V, H^S, \Theta) \cdot \widetilde{\Lambda}, \tag{5}$$

*where $\Lambda_q$ and $\Lambda_p$ are the permutation matrices for permutation $\pi_q \in \mathcal{S}^q$ and $\pi_p \in \mathcal{S}^p$, respectively. $\widetilde{\Lambda} \in \mathbb{B}^{2^n \times 2^n}$ is the unitary of permutation $\pi_{q+p}$.*

---

Appendix D.1 provides the formulation and detailed explanation of Definition 1. After measuring the quantum circuit, we can consider the model to be a mapping $\phi \in \mathbb{R}^{q+p}$. According to Definition 1, we can obtain the permutation equivariance of the mapping.

---

**Defintion 2.** *(Permutation Equivariant Mapping). A mapping $\phi(\cdot) \in \mathbb{R}^{q+p}$ is permutation equivariant for MILP graphs $(A, H^V, H^S)$ if it satisfies*

$$\phi(\Lambda_p A \Lambda_q, \Lambda_q H^V, \Lambda_p H^S) = \Lambda_{q+p} \cdot \phi(A, H^V, H^S), \tag{6}$$

*where $\Lambda_{q+p}$ is the permutation matrix representing the permutation $\pi_{q+p}$, i.e.,*

$$\Lambda_{q+p} = \begin{bmatrix} \Lambda_q & 0 \\ 0 & \Lambda_p \end{bmatrix} \in \mathbb{B}^{(q+p) \times (q+p)}.$$

---

It indicates that the mapping result after node permutation $\phi(\Lambda_p A \Lambda_q, \Lambda_q H^V)$ is equal to the result of applying the permutation $\pi_{q+p}$ to the original result $\phi(A, H^V, H^S)$. In addition, we can aggregate the result of the permutation equivariant mapping as a value to obtain the other mapping $\kappa \in \mathbb{R}$. The mapping is permutation invariant if changing the order of inputs does not alter the mapping output.

---

**Defintion 3.** *(Permutation Invariant Mapping). A mapping $\kappa(\cdot) \in \mathbb{R}$ is permutation invariant for MILP graphs if it satisfies*
$$\kappa(\Lambda_p A \Lambda_q, \Lambda_q H^V, \Lambda_p H^S) = \kappa(A, H^V, H^S).$$

---

**Theorem 1.** *In the proposed GQGLA, the circuit $U_{peqg}(A, H^V, H^S, \Theta)$ is a permutation equivariant circuit. After Pauli-Z measurement, the output of the circuit $\Phi(A, H, \Theta_{sol}) = \phi_{sol}(A, H)$ is a permutation equivariant mapping. $\phi_{fea}(A, H) = \sum_{i=1}^{q+p} \Phi(A, H, \Theta_{fea})_i$ and $\phi_{obj}(A, H) = \sum_{i=1}^{q+p} \Phi(A, H, \Theta_{obj})_i$ are permutation invariant mapping.*

---

Appendix D.2 provides a detailed proof of Theorem 1, offering theoretical guarantees of permutation equivariance and invariance for our proposed GQGLA.

### 4.4 Superior Discriminative Power over GNNs

As mentioned earlier, Chen et al. (2023) have shown that GNNs may fail to distinguish two MILP graphs, which is because of the fundamental limitations of GNNs. Xu et al. (2018) have shown that GNNs are **at most** as powerful as the Weisfeiler-Lehman (WL) test (Weisfeiler & Leman, 1968) in distinguishing graph structures. However, as the WL test iteratively updates the label of vertices in a graph only based on the label of their neighboring vertices, if two graphs exhibit subtle structural differences not reflected in the degree of vertices or the patterns of local neighbors, the WL test will fail to distinguish between these two graphs, as shown in Fig. 1. See Appendix E.1 for a detailed explanation of why GNN fails on the MILPs. By contrast, benefiting from quantum entanglement between nodes, GQGLA can capture the difference in edge connectivity between two MILP graphs. Different edges result in variations in graph-message interaction layer $U_k$, *i.e.*, $\exp(-\mathbf{i}(\sum_{(i,j)\in\mathcal{E}} A_{i,j}(I - \sigma_z)_i P_{q+j}^K))$. We give the detailed proof in Appendix E.2.

## 5 Experiments

In the experimental section, we compare the performance between classical GNNs and quantum machine learning algorithms on the learning representation of MILP graphs. The construction of the MILP dataset is in line with Chen et al. (2023) (see details in Appendix G). We use Adam (Kingma &

Table 2: Performance Comparison on the GNN-tractable MILP dataset.

| | | Feasibility (Rate of Error ↓)$/10^{-2}$ | | | Optimal Value (MSE ↓)$/10^{-2}$ | | | Optimal Solution (MSE ↓)$/10^{-1}$ | | |
|---|---|---|---|---|---|---|---|---|---|---|
| MILP-GNN (Chen et al., 2023) | # E.Size | 4 | 6 | 8 | 4 | 6 | 8 | 16 | 24 | 32 |
| | **Train** | 6.72± 0.17 | 5.42± 0.21 | 4.29± 0.12 | 1.52± 0.04 | 1.12± 0.06 | 0.66± 0.03 | 7.84± 0.21 | 6.39± 0.26 | 5.47± 0.15 |
| | **Test** | 7.96± 0.19 | 6.74± 0.20 | 5.62± 0.11 | 2.35± 0.07 | 1.64± 0.05 | 1.76± 0.04 | 8.82± 0.27 | 7.57± 0.21 | 6.31± 0.25 |
| GQGLA | # Block | 4 | 6 | 8 | 6 | 8 | 10 | 8 | 10 | 12 |
| | **Train** | 7.22± 0.11 | 6.53± 0.13 | 5.57± 0.08 | 1.10± 0.04 | 1.00± 0.02 | 0.98± 0.01 | 6.10± 0.14 | 5.79± 0.10 | 5.70± 0.09 |
| | **Test** | 7.34± 0.12 | 6.65± 0.12 | 5.74± 0.09 | 1.26± 0.05 | 1.13± 0.03 | 1.11± 0.02 | 6.37± 0.11 | 6.03± 0.08 | 5.99± 0.07 |

Ba, 2014) with an initial learning rate of 0.1 to find the optimal parameters of GQGLA, and batch size is set at 16. Experiments are performed on a single machine with 4 2.20GHz CPUs and four NVIDIA A100 GPU. The source code is written using TorchQuantum (Wang et al., 2022a), a PyTorch-based library for quantum computing, which can simulate quantum circuits with up to 26 qubits. In Table 10 of Appendix F, the performance of all possible types of GQGLA are compared, and we finally select $(W(\theta), D(\theta), K(\theta), G(\theta)) = (R_X(\theta), R_Z(\theta), R_Y(\theta), R_Z(\theta))$ for the following experiments.

## 5.1 COMPARISON WITH CLASSICAL GNNs

We first compare the GNN used in Chen et al. (2023) named MILP-GNN as it is designed for MILP. Our GQGLA model has a hyperparameter to control the number of circuit parameters, *i.e.*, the number of blocks $T$. MILP-GNN also has a hyperparameter embedding size $d$ controlling the number of parameters. We vary these two hyperparameters separately for evaluation.

**Experiments on GNN-intractable MILP Dataset.** GNN-intractable MILPs contain many pairs of WL indistinguishable graphs that cannot be distinguished by classic GNNs (Chen et al., 2023). Here, we randomly generate $2,000$ GNN-intractable MILPs with 12 variables and 6 constraints, and there are $1,000$ feasible MILPs while the others are infeasible. The error rate serves as the evaluation criterion for predicting the feasibility of GNN-intractable MILPs. We compare our GQGLA with the classical GNNs, *i.e.*, MILP-GNN (Chen et al., 2023) and MILP-GNN with random features.

MILP-GNN with random features is proposed by Chen et al. (2023) to alleviate the limitations of the original GNN by appending random features to the MILP graphs. As shown in Fig. 6, the error rate of MILP-GNN is highest regardless of the embedding size, as it cannot distinguish GNN-intractable MILPs. Although MILP-GNN with random features can improve performance, it achieves the best when the embedding size is 32, which will cost 30, 565 parameters. Moreover, adding random features may cause additional issues, *i.e.*, changing the feasibility or solution of the original problem, resulting in the change of ground truth of the dataset. In contrast, GQGLA can capture the edge connectivity of GNN-intractable MILPs, so it can achieve accurate test results with just 4 blocks, *i.e.*, 48 parameters. The results show that GQGLA has better discriminative power than GNNs with fewer parameters.

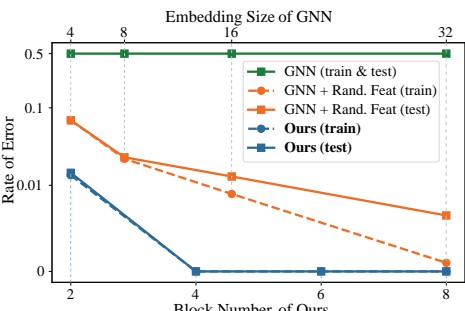

Figure 6: Comparison on GNN-intractable MILPs. *GNN* refers to MILP-GNN (Chen et al., 2023), and *GNN + Rand. feat.* indicates the MILP-GNN with random features.

**Experiments on GNN-tractable MILP Dataset.** We further compare our GQGLA and MILP-GNN on the GNN-tractable MILPs that GNNs can distinguish. We randomly generate 8, 290 GNN-tractable MILPs with four variables and four constraints, with feasible and infeasible MILPs each accounting for half. The experiments evaluate the rate of error in predicting the feasibility and the mean squared error (MSE) of predicting optimal values and optimal solutions of MILPs. As reported in Table 2, as the embedding size increases, the training error of MILP-GNN decreases, but its generalization error on the test set increases. The reason is that GNN-tractable datasets are challenging, with diverse training and test sets, making it difficult for MILP-GNN to generalize well on the test set. In contrast, our GQGLA benefits from quantum mechanisms, leading to better generalization performance.

**Comparison with Advanced GNNs.** We utilize the PyG (PyTorch Geometric) library [1] to extend more GNN models into frameworks suitable for MILP graphs. We modify each GNN to update

---
[1]https://pytorch-geometric.readthedocs.io/en/latest/

Table 4: Comparison between different quantum models on predicting the feasibility of MILPs.

| | HEA (Kandala et al., 2017) | QGCN (Zheng et al., 2021) | GQGLA (ours) |
|---|---|---|---|
| **Train** | 0.4613± 0.024 | 0.3419± 0.015 | **0.1086**± 0.008 |
| **Test** | 0.4665± 0.020 | 0.3475± 0.018 | **0.1127**± 0.012 |

Table 5: Performance changes as the number of auxiliary qubits increases on predicting optimal solution.

| # Aux. qubits | 0 | 1 | 2 | 3 |
|---|---|---|---|---|
| Train | 0.6580 | 0.6166 | **0.5694** | 0.6099 |
| Test | 0.6853 | 0.6410 | **0.5993** | 0.6354 |

variable nodes and constraint nodes using different weights. Table 3 compares our methods with GAT (Graph Attention Network) (Veličković et al., 2018), GATv2 (Brody et al., 2022), and Graph Transformer (Shi et al., 2021), *w.r.t* the accuracy of feasibility prediction on GNN-intractable and GNN-tractable datasets. As we can see, in the GNN-tractable dataset, our method remains competitive with advanced GNNs, achieving nearly the best results, and advanced GNNs still struggle with GNN-intractable datasets, whereas our model is more effective.

## 5.2 COMPARISON WITH OTHER QUANTUM ALGORITHMS

Recall in Table 1 that most quantum GNNs have not considered edge features, which yet are vital for solving MILP. Therefore, we only compare QGNNs that consider edge features, e.g. the quantum graph CNN (QGCN) (Zheng et al., 2021). We also compare a problem-agnostic, hardware-efficient ansatz (HEA) (Kandala et al., 2017). The circuit structure of HEA is typically fixed, making it unable to encode the edge information of a graph, yet can encode the node feature information of the graph. Table 4 reports the error rates on the GNN-tractable MILP dataset with three variables and three constraints. In this MILP dataset,

Table 3: Comparison of GQGLA and other GNNs on the GNN-intractable and GNN-tractable dataset.

| | Embedding Size/ # Blocks | 2 | 4 | 6 | 8 |
|---|---|---|---|---|---|
| **GNN-tractable dataset** | GAT | 0.856 | 0.926 | 0.930 | 0.940 |
| | GATv2 | 0.914 | 0.927 | 0.936 | 0.943 |
| | Graph Transformer | 0.875 | 0.914 | 0.920 | 0.930 |
| | MILP-GNN | 0.872 | 0.910 | 0.911 | 0.902 |
| | GQGLA | 0.933 | 0.936 | 0.942 | 0.946 |
| **GNN-intractable dataset** | GAT | 0.500 | 0.500 | 0.500 | 0.500 |
| | GATv2 | 0.500 | 0.500 | 0.500 | 0.500 |
| | Graph Transformer | 0.500 | 0.500 | 0.500 | 0.500 |
| | MILP-GNN | 0.500 | 0.500 | 0.500 | 0.500 |
| | GQGLA | 0.987 | 0.998 | 1.000 | 1.000 |

QGCN requires more qubits to compute than GQGLA. Moreover, we set the number of parameters for all quantum models to 96. The results show that the problem-agnostic ansatz cannot effectively learn from graph data. Although QGCN is a problem-inspired ansatz and designs an equivariant graph convolution layer, their pooling layers violate permutation invariance, leading to performance degradation in predicting MILP feasibility. By contrast, GQGLA ensures permutation invariance with better performance.

## 5.3 ABLATION STUDY

Table 5 investigates the contribution of auxiliary layers to GQGLA, which shows the performance of GQGLA in predicting the optimal solution with an increasing number of auxiliary qubits. The results indicate that increasing the number of auxiliary qubits can enhance performance, yet there may exist an optimal threshold for a specific problem scale. We can select appropriate auxiliary qubits to enhance performance for tasks of different complexity.

Furthermore, we delve into the individual components that shape GQGLA's design, as shown in Table 6. *Repeated encoding* refers to encoding features at every block. *Synchronous encoding and learning* refers to adding features and learnable parameters as new learnable parameters. *Double interaction* refers to using a constraint and a variable layer in the graph message interaction layer. All configurations employ the same number of parameters. Results indicate that these three components are useful.

Table 6: Ablation study *w.r.t* our components.

| Repeated Encoding | Syn. Encoding Learning | Double Interaction | Train | Test |
|---|---|---|---|---|
| ✗ | ✗ | ✗ | 0.0924 | 0.0953 |
| ✗ | ✗ | ✓ | 0.0876 | 0.0898 |
| ✓ | ✗ | ✓ | 0.0708 | 0.0726 |
| ✓ | ✗ | ✗ | 0.0738 | 0.0765 |
| ✓ | ✓ | ✗ | 0.0623 | 0.0659 |
| ✓ | ✓ | ✓ | 0.0557 | 0.0574 |

## 5.4 COMPARISON WITH HIGHER ORDER GNNS

We conducted experiments on BREC (Wang & Zhang, 2024) to evaluate expressiveness, compared to higher order GNNs, including subgraph-based methods NGNN (Zhang & Li, 2021), DS-GNN (Bevilacqua et al., 2022) and KP-GNN (Feng et al., 2022), k-WL hierarchy-based models $\delta$-k-LGNN (Morris et al., 2020) and PPGN (Maron et al., 2019), random model DropGNN (Papp et al., 2021), Transformer-based model Graphormer (Ying et al., 2021), and substructure-based model GSN (Bouritsas et al., 2022). The *Basic* dataset in BREC consists of

1-WL-indistinguishable graphs generated through exhaustive search and is designed to be non-regular. The *Regular* dataset contains regular graphs, further divided into simple regular graphs and strongly regular graphs, where 1-WL and 3-WL test fail, respectively. Including 4-vertex condition graphs and distance-regular graphs further increases the dataset's complexity. The *Extension* graphs bridge the gap between 1-WL

Table 7: Performance comparison of GNNs on BREC.

| Method | Basic | Regular | Extension |
|---|---|---|---|
| NGNN (Zhang & Li, 2021) | 0.983 | 0.343 | 0.59 |
| DS-GNN (Bevilacqua et al., 2022) | 0.967 | 0.343 | 1.0 |
| KP-GNN (Feng et al., 2022) | 1.0 | 0.757 | 0.98 |
| $\delta$-k-LGNN (Morris et al., 2020) | 1.0 | 0.357 | 1.0 |
| PPGN (Maron et al., 2019) | 1.0 | 0.357 | 1.0 |
| DropGNN (Papp et al., 2021) | 0.867 | 0.293 | 0.820 |
| Graphormer (Ying et al., 2021) | 0.267 | 0.086 | 0.41 |
| GSN (Bouritsas et al., 2022) | 1.0 | 0.707 | 0.95 |
| **Ours** | **1.0** | **0.964** | **1.0** |

and 3-WL, offering a more granular comparison for evaluating models beyond 1-WL. We employ Reliable Paired Comparison (RPC) (Wang & Zhang, 2024) to verify if GNNs can produce distinct outputs for a pair of graphs. Table 7 shows that the proposed GQGLA not only achieves good performance on the Basic and Extension datasets but also outperforms other classical methods on the challenging Regular dataset.

## 6 THE COMPLEXITY AND SCALABILITY OF GQGLA

For a graph with $N$ nodes and $E$ edges, the node feature encoding layer involves $O(N)$ qubits and $O(N)$ single-qubit gates, while the graph message interaction layer introduces $O(E)$ two-qubit gates. For a circuit with $T$ blocks, the total number of gates scales as $O(T(N + E))$. However, in the current NISQ era, both real quantum devices and classical simulators have a limited number of qubits. Therefore, inspired by how classical GNN algorithms handle large graph computations with limited resources, we have presented S-GQGLA to process large graphs with limited qubits. Specifically, we first employ the graph sampling technique GraphSAINT (Zeng et al., 2020) to extract appropriately connected subgraphs, then apply our GQGLA model to these subgraphs and combine the obtained information of these subgraphs together so that the training process overall learns information of the full graph. In this way, for a graph with $N$ nodes, we can sample $m$ subgraphs for training, with each subgraph using at most $k$ qubits.

Table 8 shows the performance of GQGLA on three commonly used graph classification datasets, PROTEINS and PTC. Among them, the number of nodes is up to 620 at most. We extract appropriately connected subgraphs with at most 14 nodes, which can be processed using our 14-qubit GQGLA with 6 blocks. All subgraphs are input as a batch into the GQGLA, and we combine their results to predict the classification of the full graph. We compare S-GQGLA with message-passing based GNNs GraphSAGE (Hamilton et al., 2017), GIN (Xu et al., 2018), and GAT (Veličković et al., 2018), subgraph-based methods PPGN (Maron et al., 2019), QS-CNN (Zhang et al., 2019) and Deep-WL-SGN (Xuan et al., 2019), and Transformer-based

Table 8: Accuracy comparison of S-GQGLA on three graph classification datasets.

| Methods | PROTEINS | PTC |
|---|---|---|
| GraphSAGE | 75.9 ± 3.2 | 63.9 ± 7.7 |
| GIN | 76.20 ± 2.8 | 64.6 ± 7.0 |
| GAT | 74.70 ± 2.2 | 66.70 ± 5.1 |
| PPGN | 76.66 ± 5.6 | 62.94 ± 6.6 |
| QS-CNN | 78.2 ± 4.6 | 66.0 ± 4.4 |
| Deep-WL-SGN | 76.78 ± 2.4 | 65.88 ± 5.1 |
| U2GNN | 78.53 ± 4.1 | 69.63 ± 3.6 |
| SEG-BERT | 77.20 ± 3.1 | 68.86 ± 4.2 |
| **S-GQGLA (Ours)** | 91.64 ± 3.1 | 68.57 ± 2.9 |

methods SEG-BERT (Zhang, 2020) and U2GNN (Nguyen et al., 2022a). S-GQGLA can achieve the best performance on PROTEINS, and slightly worse than U2GNN and SEG-BERT on PTC.

## 7 CONCLUSION

This paper introduces an expressive quantum graph learning framework aimed at addressing complex graph structural data, such as learning the MILP graph representation. Our proposed framework, GQGLA, uses the synchronous encoding and learning module to handle node and edge features, and presents a parameter-sharing mechanism and a carefully designed graph information interaction layer. We theoretically prove that GQGLA possesses permutation equivariance and invariance for nodes and edges. Moreover, the proposed graph information interaction layer leverages quantum entanglement to model the edges, enabling the capture of graph patterns that are challenging for classical methods. Numerical experiments demonstrate that our approach can overcome the fundamental limitations of traditional GNNs, achieving superior performance on MILP tasks that include GNN-intractable graphs. GQGLA also offers flexibility in configuration for general graphs, and the results of the BREC dataset highlight the generality and separating power of GQGLA.

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

# A RELATED WORK

**Quantum Graph Neural Networks** Various quantum graph neural networks have been proposed, with applications in fields such as social networks (Ye et al., 2023b) and molecular generation (Li & Ghosh, 2022; Wu et al., 2024a; Yang et al., 2025). Verdon et al. (2019) proposed a class of graph neural networks by defining operations in terms of Hamiltonians based on the graph structure. However, their models are restricted to Hamiltonians of specific forms, thereby cannot flexibly and efficiently encode classical high-dimensional node or edge features of the graphs to solve some classical tasks. Zheng et al. (2021) designed a specific quantum graph convolutional neural network (QGCN), which uses an amplitude encoding method to encode node and edge features and employs qubits representing edges as control qubits to apply unitaries to the two qubits representing nodes connected by that edge. Nevertheless, the usage of edge qubits will lead to the number of qubits of the model scales quadratically with the number of nodes. Moreover, the pooling layer and measurement operator of QGCN will indeed result in the loss of permutation invariance of the entire model. Ai et al. (2022) presented an ego-graph based Quantum Graph Neural Network (egoQGNN), which decomposes the input graph into smaller-scale subgraphs and feeds them into the circuit. However, due to the use of entanglement layers within the model, it still does not possess permutation invariance.

**Equivariant Quantum Neural Networks** Recently, a nascent field named geometric quantum machine learning (GQML) (Larocca et al., 2022; Nguyen et al., 2022b) has been developed, which leverages the machinery of group and representation theory (Ragone et al., 2022) to build quantum architectures that encode symmetry information about the problem. Schatzki et al. (2022) provide an analytical study of $S_n$-equivariant QNNs and prove that they do not suffer from barren plateaus, quickly reach overparametrization, and can generalize well from small amounts of data. The equivariant QNNs can used to learn various problems with permutation symmetries abound, such as molecular systems, condensed matter systems, and distributed quantum sensors (Peruzzo et al., 2014; Guo et al., 2020; Wu et al., 2024b), namely, they are also not specifically designed to solve classical graph tasks. Mernyei et al. (2022) first proposed a theoretical recipe for building permutation equivariant quantum graph circuits (EQGC) and aggregated the output of the quantum circuit by classical functions to ensure permutation invariance of the model. Nevertheless, the EQGC does not provide the specific circuit implementation and does not consider the case of weighted graphs in their model. In addition, another QNN with permutation equivariance Ye et al. (2023a) is proposed, which is specially designed for solving quadratic assignment problems, but their model only encodes the graph information and then employs the shared problem-agnostic ansatz to learn the representation of each node. Thus, their model does not contain the learnable graph message interaction layer.

**Quantum Algorithms for MILP** Mixed-Integer Linear Programming (MILP) is a mathematical optimization approach that aims to find the best solution to a linear objective function while imposing constraints on some or all of the variables to be integers. MILP is widely used in various practical applications such as process scheduling (Floudas & Lin, 2005a; Li et al., 2023b; 2024; Wu et al., 2025b), transportation (Richards & How, 2002; Wu et al., 2025a; Bao et al., 2024; 2025), and network design (Fortz & Poss, 2009; Li et al., 2023a). Recently, researchers have endeavored to employ quantum computing to assist in solving the MILP. Zhao et al. (2022) proposed a hybrid quantum-classical Benders' decomposition algorithm, which decomposes an MILP problem into a Quadratic unconstrained binary optimization (QUBO) problem solved by quantum computer and a subproblem easily tackled by classical computers. Ossorio & Pena (2022) described an algorithm based on Dantzig–Wolfe decomposition. Different from Zhao et al. (2022), the algorithm then solves several either continuous or binary subproblems instead of a mixed one. Wang et al. (2022b) pointed out that quantum-inspired Ising machines can be used to solve MILPs by reducing them into Ising models. However, the above algorithms are based on unconstrained Ising models, while MILPs are subject to complex constraints. Their common solution is to introduce a penalty to the algorithm. A proper penalty is of great importance because an extremely large penalty may cause the quantum annealer to malfunction since it will explode the coefficients, while a soft penalty may make the quantum annealer ignore the corresponding constraints (Zhao et al., 2022). However, there is no instruction on how to tune the penalty, and it may even be different for various MILP problems. In contrast, our approach leverages QML to represent MILP problems, thereby pioneering a novel direction for harnessing quantum computing in aiding MILP solutions, there is promising for witnessing the emergence of new paradigms that combine quantum and classical methods for MILP solving.

## B  THE BASICS OF QUANTUM COMPUTING

**Single-qubit Quantum State.** In quantum computing, the fundamental building blocks of computation are qubits (short for quantum bits), which are the quantum analog of classical bits. Unlike classical bits, which can only take on one of two possible values (0 or 1), a qubit can exist in a superposition of the two states, represented by the vector:

$$|\psi\rangle = \alpha_1|0\rangle + \alpha_2|1\rangle, \tag{7}$$

where $|0\rangle$ and $|1\rangle$ represent the two basis states of one qubit, and $\alpha_1$ and $\alpha_2$ are complex numbers that satisfy the normalization condition $|\alpha_1|^2 + |\alpha_2|^2 = 1$. When $|\psi\rangle$ is measured, it will collapse to either the $|0\rangle$ or $|1\rangle$ state with a probability $|\alpha_1|^2$ or $|\alpha_2|^2$.

Mathematically, the quantum state of one qubit can be denoted as a complex 2-dimensional vector, e.g., $|0\rangle = [1,0]^T$, $|1\rangle = [0,1]^T$, and $|\psi\rangle = [\alpha_1, \alpha_2]^T$. The Bloch sphere is a sphere of radius 1, which is a useful tool for visualizing the state of a single qubit. Any other state of one qubit can be represented by a point on the surface of the sphere, and you can see them through this online tool [2].

**Multi-qubit Quantum State.** Multi-qubit quantum states are an extension of single-qubit quantum states, and a $N$-qubit quantum state can be represented as a complex $2^N$-dimensional vector in Hilbert space. This is why quantum systems are often described as living in a $2^N$-dimensional Hilbert space. More specifically, a two-qubit system can be represented as $|\phi\rangle = \alpha_1|00\rangle + \alpha_2|01\rangle + \alpha_3|10\rangle + \alpha_4|11\rangle$, where $\sum_{i=1}^{2^2} |\alpha_i^2| = 1$ and $|00\rangle$ represent the tensor product $|0\rangle \otimes |0\rangle$.

**Quantum Circuits.** Quantum circuits are constructed using quantum gates, which are analogous to classical logic gates. Some commonly used single-qubit gates include the Pauli-X gate, the Pauli-Y gate, and the Pauli-Z gate. They can be represented by the unitary matrix:

$$\sigma_x = \begin{pmatrix} 0 & 1 \\ 1 & 0 \end{pmatrix}, \sigma_y = \begin{pmatrix} 0 & -i \\ i & 0 \end{pmatrix}, \sigma_z = \begin{pmatrix} 1 & 0 \\ 0 & -1 \end{pmatrix}. \tag{8}$$

The Controlled-NOT (CNOT) gate is a two-qubit gate that flips the second qubit (target) if the first qubit (control) is in the $|1\rangle$ state. We provide the matrix forms of common quantum gates in Table 9. When a quantum gate acts on a quantum state $|\psi\rangle$, it transforms this state to another quantum state $|\psi'\rangle$, according to the mathematical operation $|\psi'\rangle = U|\psi\rangle$, where $U$ represents the unitary matrix associated with the quantum gate.

**Parameterized Quantum Circuits.** Parameterized quantum circuits (PQCs) consist of parameterized gates and offer a concrete way to implement quantum machine learning algorithms. Specifically, the common parameterized quantum gates are listed in Table 9. The parameters (e.g., $\theta$) in the quantum gate can be either learnable parameters for optimizers or classical information that we want to encode.

**Quantum Machine Learning.** A quantum machine learning model can be constructed using a sequence of parameterized quantum gates. The initial quantum states can be transformed into the output quantum states. By measuring the output of the quantum circuit, we can convert quantum information into classical information, which can be used to calculate the cost function of the optimization task. We can use classical optimizers to minimize the cost function by adjusting the parameters of quantum gates.

## C  FORMULAS OF EACH LAYER IN GQGLA

In GQGLA, the single-qubit gate can be written as the unitary matrix from. For example, $K(\theta) = \exp(-\mathbf{i}\frac{\theta}{2}P^K)$, where $P^K \in \{\sigma_x, \sigma_y, \sigma_z\}$. Similarly, $W(\theta) = \exp(-\mathbf{i}\frac{\theta}{2}P^W)$, $D(\theta) = \exp(-\mathbf{i}\frac{\theta}{2}P^D)$, $G(\theta) = \exp(-\mathbf{i}\frac{\theta}{2}P^G)$.

Based on this, the unitary matrix of the node feature layer of Fig. 4 contains:
$$U_W(c, u, b, \alpha_t) = \exp(-\mathbf{i}(\sum_{i=1}^{q}((c_i + \alpha_{t,1})P_{2i}^W + (u_i + \alpha_{t,2})P_{2i+1}^W) + \sum_{j=1}^{p}(b_j + \alpha_{t,5})P_{2q+j}^W)),$$
$$U_D(l, \epsilon, \circ, \alpha_t) = \exp(-\mathbf{i}(\sum_{i=1}^{q}((l_i + \alpha_{t,3})P_{2i}^D + (\epsilon_i + \alpha_{t,4})P_{2i+1}^D) + \sum_{j=1}^{p}(\circ_j + \alpha_{t,6})P_{2q+j}^D)),$$
$$\text{where } P_{2i}^W = \underbrace{I \otimes \cdots \otimes}_{2i-1} P^W \underbrace{\otimes \cdots \otimes I}_{q+p-2i}. \tag{9}$$

---

[2]https://javafxpert.github.io/grok-bloch/

Table 9: Common quantum gates.

| Operator | Gate(s) | Matrix |
|---|---|---|
| Pauli-X (X, $\sigma_x$) | $\boxed{X}$ | $\begin{bmatrix} 0 & 1 \\ 1 & 0 \end{bmatrix}$ |
| Pauli-Y (Y, $\sigma_y$) | $\boxed{Y}$ | $\begin{bmatrix} 0 & -i \\ i & 0 \end{bmatrix}$ |
| Pauli-Z (Z, $\sigma_z$) | $\boxed{Z}$ | $\begin{bmatrix} 1 & 0 \\ 0 & -1 \end{bmatrix}$ |
| Rotation-Z ($R_Z(\theta)$) | $\boxed{RZ}$ | $\begin{bmatrix} e^{-\mathbf{i}\frac{\theta}{2}} & 0 \\ 0 & e^{\mathbf{i}\frac{\theta}{2}} \end{bmatrix}$ |
| Rotation-Y ($R_Y(\theta)$) | $\boxed{RY}$ | $\begin{bmatrix} \cos(\frac{\theta}{2}) & -\sin(\frac{\theta}{2}) \\ \sin(\frac{\theta}{2}) & \cos(\frac{\theta}{2}) \end{bmatrix}$ |
| Rotation-X ($R_X(\theta)$) | $\boxed{RX}$ | $\begin{bmatrix} \cos(\frac{\theta}{2}) & -\mathbf{i}\sin(\frac{\theta}{2}) \\ -\mathbf{i}\sin(\frac{\theta}{2}) & \cos(\frac{\theta}{2}) \end{bmatrix}$ |
| Controlled Not (CNOT, CX) | ●—⊕ | $\begin{bmatrix} 1 & 0 & 0 & 0 \\ 0 & 1 & 0 & 0 \\ 0 & 0 & 0 & 1 \\ 0 & 0 & 1 & 0 \end{bmatrix}$ |
| Controlled RZ ($C_{RZ}(\theta)$) | ●—$\boxed{RZ}$ | $\begin{bmatrix} 1 & 0 & 0 & 0 \\ 0 & 1 & 0 & 0 \\ 0 & 0 & e^{-\mathbf{i}\frac{\theta}{2}} & 0 \\ 0 & 0 & 0 & e^{\mathbf{i}\frac{\theta}{2}} \end{bmatrix}$ |
| Controlled RY ($C_{RY}(\theta)$) | ●—$\boxed{RY}$ | $\begin{bmatrix} 1 & 0 & 0 & 0 \\ 0 & 1 & 0 & 0 \\ 0 & 0 & \cos(\frac{\theta}{2}) & -\sin(\frac{\theta}{2}) \\ 0 & 0 & \sin(\frac{\theta}{2}) & \cos(\frac{\theta}{2}) \end{bmatrix}$ |
| Controlled RX ($C_{RX}(\theta)$) | ●—$\boxed{RX}$ | $\begin{bmatrix} 1 & 0 & 0 & 0 \\ 0 & 1 & 0 & 0 \\ 0 & 0 & \cos(\frac{\theta}{2}) & -\mathbf{i}\sin(\frac{\theta}{2}) \\ 0 & 0 & -\mathbf{i}\sin(\frac{\theta}{2}) & \cos(\frac{\theta}{2}) \end{bmatrix}$ |

The graph message interaction layer of Fig. 5 contains:

$$
\begin{aligned}
U_K^V(A, \beta_{t,1}) &= \exp(-\mathbf{i}(\sum_{(i,j)\in\mathcal{E}} (A_{i,j} + \beta_{t,1})(I - \sigma_z)_{2i} P_{2q+j}^K)), \\
U_K^{V'}(A, \beta_{t,2}) &= \exp(-\mathbf{i}(\sum_{(i,j)\in\mathcal{E}} (A_{i,j} + \beta_{t,2})(I - \sigma_z)_{2i+1} P_{2q+j}^K)), \\
U_G^V(\beta_{t,3}) &= \exp(-\mathbf{i}(\sum_{i} \beta_{t,3}(I - \sigma_z)_{2i+1} P_{2i}^G)), \\
U_K^S(A, \beta_{t,4}) &= \exp(-\mathbf{i}(\sum_{(i,j)\in\mathcal{E}} (A_{i,j} + \beta_{t,4})(I - \sigma_z)_{2q+j} P_{2i}^K)), \\
U_K^{S'}(A, \beta_{t,5}) &= \exp(-\mathbf{i}(\sum_{(i,j)\in\mathcal{E}} (A_{i,j} + \beta_{t,5})(I - \sigma_z)_{2q+j} P_{2i+1}^K)), \\
U_G^S(\beta_{t,6}) &= \exp(-\mathbf{i}(\sum_{j} \beta_{t,6} P_{2q+j}^G)).
\end{aligned}
\tag{10}
$$

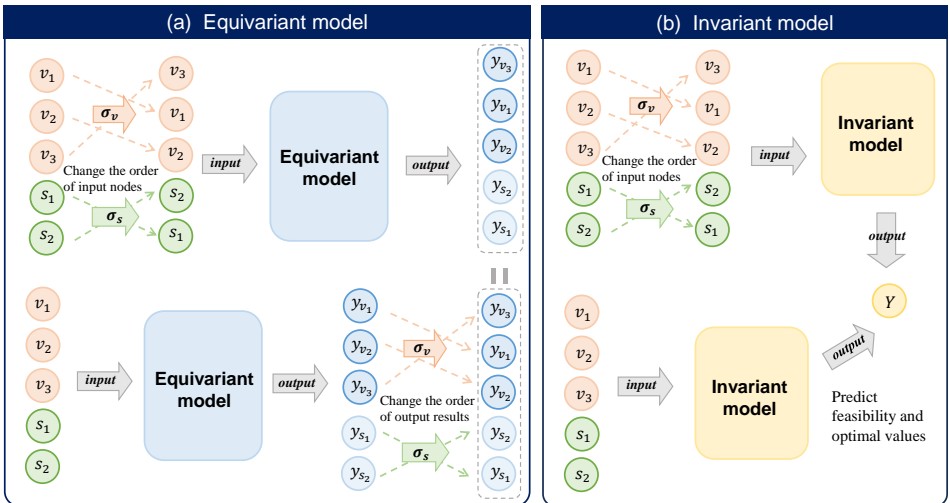

Figure 7: Diagram of the properties of equivariant (a) and invariant (b) models. For equivariant models, when the input node has a permutation $\sigma$, the output is equivalent to the original one with the same permutation $\sigma$. For invariant models, the permutation of input nodes will not affect output $Y$.

# D  THEOREMS AND PROOFS OF THE EQUIVARIANCE

## D.1  THE DETAILS OF DEFINITION 1

$\Lambda_q \in \mathbb{B}^{q \times q}$ and $\Lambda_p \in \mathbb{B}^{p \times p}$ are permutation matrices representing a permutation $\pi_q \in \mathcal{S}^q$ and $\pi_p \in \mathcal{S}^p$, respectively. $\mathcal{S}^q$ and $\mathcal{S}^p$ are the groups containing all permutations on the $q$ variable nodes and $p$ constraint nodes, respectively. After the MILP graph undergoes permutation $\pi_q$ and $\pi_p$, the feature set of the variable nodes and the feature set of the constraint nodes become $\Lambda_q H^V$ and $\Lambda_p H^S$, respectively. Then, the adjacency matrix between variable nodes and constraint nodes of the MILP graph $A \in \mathbb{R}^{p \times q}$ is transformed to $\Lambda_p A \Lambda_q$. We define a permutation $\pi_{q+p}$ on the $(q+p)$ elements to indicate the overall permutation on all nodes of the MILP graph. $\pi_{q+p}$ consists of two distinct parts: the first $q$ elements follow the permutation $\pi_q$, and the subsequent $p$ elements follow $\pi_p$. $\widetilde{\Lambda}$ is the unitary representation of $\pi_{q+p}$, which indicates the permutation of qubits representing nodes. $\widetilde{\Lambda}$ can be implemented by applying a series of $SWAP$ gates to the quantum circuit. The detailed proof and formulation are as follows.

It is known that any permutation can be expressed as a product of transpositions. The transposition refers to a simple permutation that just swaps two elements. Suppose that the permutation $\pi_{q+p}$ contains $z$ transpositions, i.e., $\pi_{q+p} = (\delta_{11}, \delta_{12})...(\delta_{i1}, \delta_{i2})...(\delta_{z1}, \delta_{z2})$. Given that a $n$-qubit quantum circuit and suppose that one qubit represents one node, i.e., $n = p + q$, a transposition $(\delta_{z1}, \delta_{z2})$ can be represented by a $SWAP_{(\delta_{z1}, \delta_{z2})}$ gate to exchange the $\delta_{z1}$-th qubit and $\delta_{z2}$-th qubit of the quantum circuit. Hence, the corresponding unitary $\widetilde{\Lambda} = \widetilde{\Lambda}_{\delta_1} \widetilde{\Lambda}_{\delta_2}...\widetilde{\Lambda}_{\delta_i}...\widetilde{\Lambda}_{\delta_z}$ is equal to

$$SWAP_{(\delta_{11}, \delta_{12})}...SWAP_{(\delta_{i1}, \delta_{i2})}...SWAP_{(\delta_{z1}, \delta_{z2})}. \tag{11}$$

That is, we decompose a complex permutation into a series of transposition, so that we can get:

**Defintion D.1.** *If the quantum circuit $U$ is equivariant for any transpositions $(\delta_{i1}, \delta_{i2})$, i.e.,*

$$U(\Lambda_{p\delta_i} A \Lambda_{q\delta_i}, \Lambda_{q\delta_i} H^V, \Lambda_{p\delta_i} H^S, \Theta) = \widetilde{\Lambda}_{\delta_i}^{\dagger} \cdot U(A, H^V, H^S, \Theta) \cdot \widetilde{\Lambda}_{\delta_i},$$

*the circuit is equivariant for the permutation $\pi_{q+p}$, where $\Lambda_{p\delta_i}$ and $\Lambda_{q\delta_i}$ are the permutation matrices corresponding to the permutation $(\delta_{i1}, \delta_{i2})$.*

Based on Definition D.1, we can derive Definition 1 by using Eq. 11.

$$
\widetilde{\Lambda}^\dagger \cdot U(A, \mathrm{H^V}, \mathrm{H^S}, \Theta) \cdot \widetilde{\Lambda}
$$
$$
= SWAP_{(\delta_{z1}, \delta_{z2})}...SWAP_{(\delta_{11}, \delta_{12})} \cdot U(A, \mathrm{H^V}, \mathrm{H^S}, \Theta) \cdot SWAP_{(\delta_{11}, \delta_{12})}...SWAP_{(\delta_{z1}, \delta_{z2})},
$$
$$
= SWAP_{(\delta_{z1}, \delta_{z2})}...SWAP_{(\delta_{21}, \delta_{22})} U(\Lambda_{p\delta_1} A \Lambda_{q\delta_1}, \Lambda_{q\delta_1} \mathrm{H^V}, \Lambda_{p\delta_1} \mathrm{H^S}, \Theta) SWAP_{(\delta_{21}, \delta_{22})}...SWAP_{(\delta_{z1}, \delta_{z2})}
$$
$$
= U(\Lambda_{p\delta_1...\delta_z} A \Lambda_{q\delta_1...\delta_z}, \Lambda_{q\delta_1...\delta_z} \mathrm{H^V}, \Lambda_{p\delta_1...\delta_z} \mathrm{H^S}, \Theta)
$$
$$
= U(\Lambda_p A \Lambda_q, \Lambda_q \mathrm{H^V}, \Lambda_p \mathrm{H^S}, \Theta)
$$

(12)

By the associative law of matrix multiplication, we can combine the middle three terms of the formula until the last transpositions. In the final, $\Lambda_{p\delta_1...\delta_z} = \Lambda_p$ and $\Lambda_{q\delta_1...\delta_z} = \Lambda_q$ reprsent the permutation matrices corresponding to the permutation $\pi_{q+p} = (\delta_{11}, \delta_{12})...(\delta_{i1}, \delta_{i2})...(\delta_{z1}, \delta_{z2})$. Thereby, we can obtain Definition 1.

## D.2    THE DETAILS OF THEOREM 1

We can decompose Theorem 1 into the following theorems and corollary for clearer proof.

**Theorem D.1.** *( $U_{qgl}$ is Equivariant). The circuit $U_{qgl}(A, H^V, H^S, \Theta)$ of the proposed GQGLA is a permutation equivariant circuit.*

**Theorem D.2.** *( $\Phi(A, H, \Theta)$ is an Equivariant Mapping). After Pauli-Z measurement, the output of the GQGLA is permutation equivariant.*

**Corollary D.1.** *( $\phi(A, H)_{fea}$ is an Invariant Mapping). The mapping of GQGLA to predict the feasibility of instance $\phi(A, H)_{fea}$ is permutation invariant.*

### D.2.1    THE PROOF OF THEOREM D.1

These two circuits in the quantum circuit are permutation-dependent and permutation-independent circuits. Permutation-dependent circuit refers to the quantum circuit composed of gates related to input node features or edge features. Permutation independent circuit is defined as:

**Defintion D.2.** *(Permutation Independent Circuit). A parameterized quantum circuit $U_{ind}$) is permutation independent for MILP graphs $(A, H^V, H^S)$ if $U_{ind}(\Lambda_p A \Lambda_q, \Lambda_q H^V, \Lambda_p H^S, \Theta) = U_{ind}(\Theta)$, and*

$$
\widetilde{\Lambda}^\dagger U_{ind}(\Theta) = U_{ind}(\Theta)\widetilde{\Lambda} = U_{ind}(\Theta).
$$
(13)

We can derive the permutation equivariance of the circuit that only contain permutation equivariant circuit and permutation indepentent circuit.

**Corollary D.2.** *( Equivariant $\times$ Independent = Equivariant). If a quantum circuit $U = U_{eq}U_{ind}$, where $U_{eq}$ represents the permutation equivariant circuit and $U_{ind}$ represents the permutation independent circuit, then $U$ is the permutation equivariant circuit.*

*Proof.* It is known that

$$
U_{eq}(A, \mathrm{H^V}, \mathrm{H^S}) = \widetilde{\Lambda} \cdot U_{eq}(\Lambda_p A \Lambda_q, \Lambda_q \mathrm{H^V}, \Lambda_p \mathrm{H^S}) \cdot \widetilde{\Lambda}^\dagger,
$$
$$
\widetilde{\Lambda}^\dagger U_{ind}(\Theta) = U_{ind}(\Theta)\widetilde{\Lambda}^\dagger = U_{ind}(\Theta).
$$

Therefore, we can obtain

$$
U(A, \mathrm{H^V}, \mathrm{H^S}, \Theta) = U_{eq}(A, \mathrm{H^V}, \mathrm{H^S})U_{indd}(\Theta) = \widetilde{\Lambda} \cdot U_{eq}(\Lambda_p A \Lambda_q, \Lambda_q \mathrm{H^V}, \Lambda_p \mathrm{H^S}) \cdot \widetilde{\Lambda}^\dagger \cdot U_{ind}(\Theta)
$$
$$
= \widetilde{\Lambda} \cdot U_{eq}(\Lambda_p A \Lambda_q, \Lambda_q \mathrm{H^V}, \Lambda_p \mathrm{H^S}) \cdot \widetilde{\Lambda}^\dagger \cdot \widetilde{\Lambda} \cdot U_{ind}(\Theta) \cdot \widetilde{\Lambda}^\dagger
$$
$$
= \widetilde{\Lambda} \cdot U_{eq}(\Lambda_p A \Lambda_q, \Lambda_q \mathrm{H^V}, \Lambda_p \mathrm{H^S}) \cdot U_{ind}(\Theta) \cdot \widetilde{\Lambda}^\dagger
$$
$$
= \widetilde{\Lambda} \cdot U(\Lambda_p A \Lambda_q, \Lambda_q \mathrm{H^V}, \Lambda_p \mathrm{H^S}, \Theta) \cdot \widetilde{\Lambda}^\dagger,
$$

which indicates the circuit satisfies the definition of permutation equivariance.

On the other hand, GQGLA contains many layers, and we can demonstrate the permutation equivariance of GQGLA by proving the permutation equivariance of each layer in GQGLA.

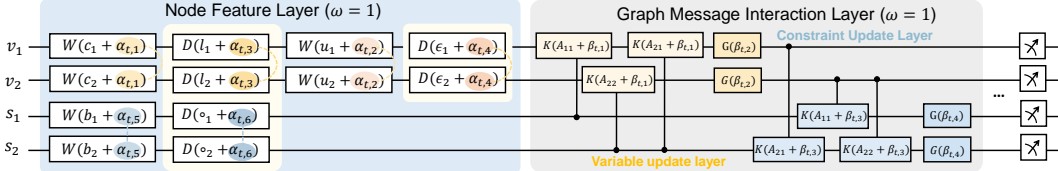

Figure 8: The framework of GQGLA with one qubit represents one node (*i.e.*, $\omega = 1$).

**Theorem D.3.** *(Equivariance of $\Gamma$-layered circuit).* A $\Gamma$-layered quantum circuit $U(A, H^V, H^S, \Theta) = \prod_{t=1}^{\Gamma} U^t(A, H^V, H^S, \Theta_t)$ is permutation equivariant iff every layer $U^t$ is permutation equivariant.

*Proof.* $U^t$ is permutation equivariant means that

$$U^t(A, \mathrm{H}^V, \mathrm{H}^S, \Theta_t) = \widetilde{\Lambda} \cdot U^t(\Lambda_p A \Lambda_q, \Lambda_q \mathrm{H}^V, \Lambda_p \mathrm{H}^S, \Theta_t) \cdot \widetilde{\Lambda}^\dagger.$$

Therefore,

$$U(A, \mathrm{H}^V, \mathrm{H}^S, \Theta) = \prod_{t=1}^{\Gamma} U^t(A, \mathrm{H}^V, \mathrm{H}^S, \Theta_t) = \prod_{t=1}^{\Gamma} \widetilde{\Lambda} \cdot U^\top(\Lambda_p A \Lambda_q, \Lambda_q \mathrm{H}^V, \Lambda_p \mathrm{H}^S, \Theta_t) \cdot \widetilde{\Lambda}^\dagger,$$

where $\widetilde{\Lambda}$ and $\widetilde{\Lambda}^\dagger \widetilde{\Lambda} = I$. Hence,

$$U(A, \mathrm{H}^V, \mathrm{H}^S, \Theta) = \widetilde{\Lambda}(\prod_{t=1}^{\Gamma} U^(\Lambda_p A \Lambda_q, \Lambda_q \mathrm{H}^V, \Lambda_p \mathrm{H}^S, \Theta_t))\widetilde{\Lambda}^\dagger = \widetilde{\Lambda} U(\Lambda_p A \Lambda_q, \Lambda_q \mathrm{H}^V, \Lambda_p \mathrm{H}^S, \Theta)\widetilde{\Lambda}^\dagger.$$

Each layer of GQGLA has the same circuit structure and can be decomposed into $U_x(\mathrm{H}^V, \mathrm{H}^S, \alpha_t) U_g(A, \beta_t) U_a(\gamma_t)$. Moreover, $U_a(\gamma_t)$ is a permutation independent circuit. Therefore, by Corollary D.2 and Corollary D.1, we only need to prove that $U_x(\mathrm{H}^V, \mathrm{H}^S, \alpha_t)$ and $U_g(A, \beta_t)$ are permutation equivariant for any transpositions $(\delta_{i1}, \delta_{i2})$.

**Equivariance of the Node Feature Layer**

Take Fig. 8 as an example, *i.e.*, one qubit represents one node, $U_x(\mathrm{H}^V, \mathrm{H}^S, \alpha_t)$ can be written as $U_{v_1} \otimes U_{v_2} \otimes U_{s_1} \otimes U_{s_2}$, where $U_{v_i} = W(c_i + \alpha_{t,1}) D(l_i + \alpha_{t,3}) W(u_i + \alpha_{t,2}) D(\epsilon_i + \alpha_{t,4}), i = \{1, ..., q\}$ and $U_{s_j} = W(b_j + \alpha_{t,5}) D(\circ_j + \alpha_{t,6}), j = \{1, ..., p\}$. When the node permutation changes, $U_x(\Lambda_q \mathrm{H}^V, \Lambda_p \mathrm{H}^S, \alpha_t) = U_{v_{\pi_q(1)}} \otimes U_{v_{\pi_q(2)}} \otimes U_{s_{\pi_p(1)}} \otimes U_{s_{\pi_p(2)}}$. As we can see, the node permutation is transformed into the permutation of the order of tensor products of unitary matrices on individual qubits. Suppose there is a $n$-qubit arbitrary quantum state

$$|\psi\rangle = \sum C_{d_1...d_j...d_n} |d_1...d_j...d_n\rangle, d_j \in \{0, 1\},$$

where $C_{d_1...d_j...d_n}$ is the amplitude of the basic state $|d_1...d_j...d_n\rangle$. Take $|\psi\rangle$ as input to $U = U_1 \otimes ...U_j... \otimes U_n$. For a transposition $(\delta_{i1}, \delta_{i2})$,

$$SWAP_{(\delta_{i1}, \delta_{i2})}|\psi\rangle = \sum C_{d_1...d_{\delta_{i1}}...d_{\delta_{i2}}...d_n} |d_1...d_{\delta_{i2}}...d_{\delta_{i1}}...d_n\rangle,$$

$$USWAP_{(\delta_{i1}, \delta_{i2})}|\psi\rangle = \sum C_{d_1...d_{\delta_{i1}}...d_{\delta_{i2}}...d_n} U_1|d_1\rangle \otimes ... \otimes U_{\delta_{i1}}|d_{\delta_{i2}}\rangle \otimes ...U_{\delta_{i2}}|d_{\delta_{i1}}\rangle \otimes ...U_n|d_n\rangle,$$

$$SWAP_{(\delta_{i1}, \delta_{i2})} USWAP_{(\delta_{i1}, \delta_{i2})}|\psi\rangle = \sum C_{d_1...d_{\delta_{i1}}...d_{\delta_{i2}}...d_n} U_1|d_1\rangle \otimes ... \otimes U_{\delta_{i2}}|d_{\delta_{i1}}\rangle \otimes ...U_{\delta_{i1}}|d_{\delta_{i2}}\rangle \otimes ...U_n|d_n\rangle$$

$$= U_1 \otimes ... \otimes U_{\delta_{i2}} \otimes ... \otimes U_{\delta_{i1}} \otimes ... \otimes U_n|\psi\rangle. \tag{14}$$

Therefore, we can obtain $SWAP_{(\delta_{i1}, \delta_{i2})} U_1 \otimes ... \otimes U_{\delta_{i1}} \otimes ... \otimes U_{\delta_{i2}} \otimes ... \otimes U_n SWAP_{(\delta_{i1}, \delta_{i2})} = U_1 \otimes ... \otimes U_{\delta_{i2}} \otimes ... \otimes U_{\delta_{i1}} \otimes ... \otimes U_n$, which indicates the unitary $U$ satisfies the definition of the equivariance.

**Equivariance of the Graph Interaction Layer**

As shown in Fig. 8, the variable update layer has a similar structure to the constraint update layer. Moreover, the layer that consists of $G(\beta_t)$ is the permutation independent circuit. Therefore, according to Corollary D.2, we only need to prove that the layer that consists of controlled-$K$ gates is

permutation equivariant. It is known that $CK(\theta)_{2,1} = I \otimes |0\rangle\langle0| + K(\theta) \otimes |1\rangle\langle1|$, where the first qubit is the target qubit and the second qubit is the control qubit. Moreover,

$$SWAP_{(2,1)}CK(\theta)_{2,1}SWAP_{(2,1)} = |0\rangle\langle0| \otimes I + |1\rangle\langle1| \otimes K(\theta) = CK(\theta)_{1,2}. \tag{15}$$

For $n(= q + p)$-qubit circuit,

$$CK(A_{jk} + \beta_t)_{q+j,k} = I_1 \otimes ... \otimes |0\rangle\langle0|_{q+j} \otimes ... \otimes I_n + I_1 \otimes K(A_{jk} + \beta_t)_k \otimes ... \otimes |1\rangle\langle1|_{q+j} \otimes ... \otimes I_n,$$

where the index of the matrix represents the index of qubit that the matrix acted on. For brevity, we set $\theta_{jk} = A_{jk} + \beta_t$, $\mathbf{0} = |0\rangle\langle0|$, $\mathbf{1} = |1\rangle\langle1|$, and the edge set of the graph is $\mathcal{E}$. As we can see, the double-qubit quantum gate is only related to the permutation of the qubit $(q + j)$ or qubit $k$. Assume that there are two transpositions $(q + j, q + \delta^s(j))$ and $(k, )$ to exchange the place of the qubits, and $SWAP'$ denot Considering a $n$-qubit arbitrary quantum state

$$|\psi\rangle = \sum C_{d_1...d_i...d_n}|d_1...d_k...d_{\delta^v(k)}...d_{q+j}...d_{q+\delta^s(j)}...d_n\rangle, d_i \in \{0,1\}.$$

$$SWAP'|\psi\rangle = \sum C_{d_1...d_i...d_n}|d_1...d_{\delta^v(k)}...d_k...d_{q+\delta^s(j)}...d_{q+j}...d_n\rangle$$

$$CK(\theta_{jk})_{q+j,k}SWAP'|\psi\rangle = \sum C_{d_1...d_i...d_n}((|d_1\rangle \otimes ... \otimes \mathbf{0}d_{q+\delta^s(j)} \otimes ... \otimes |d_{q+j}\rangle \otimes ... \otimes |d_{\pi}(n)\rangle))$$

$$+ (|d_1\rangle \otimes ... \otimes K(\theta_{jk})|d_{\delta^v(k)}\rangle \otimes ... \otimes |d_k\rangle \otimes ... \otimes \mathbf{1}d_{q+\delta^s(j)}\rangle \otimes ... \otimes |d_{q+j}\rangle \otimes ... \otimes |d_{\pi}(n)\rangle\rangle,$$

$$SWAP'^T CK(\theta_{jk})_{q+j,k}SWAP'|\psi\rangle = \sum C_{d_1...d_i...d_n}((|d_1\rangle \otimes ... \otimes d_{q+j} \otimes ... \otimes \mathbf{0}d_{q+\delta^s(j)}\rangle \otimes$$

$$... \otimes |d_{\pi}(n)\rangle)) + (|d_1\rangle \otimes ... \otimes |d_k\rangle \otimes ... \otimes K(\theta_{jk})|d_{\delta^v(k)}\rangle \otimes ... \otimes |d_{q+j}\rangle \otimes ... \otimes \mathbf{1}|d_{q+\delta^s(j)}\rangle \otimes ... \otimes |d_{\pi}(n)\rangle\rangle,$$

$$= CK(A_{jk} + \beta_t)_{q+\delta^s(j),\delta^v(k)}.|\psi\rangle \tag{16}$$

Therefore, we can obtain

$$SWAP'^T CK(A_{jk} + \beta_t)_{q+j,k}SWAP'|\psi\rangle = CK(A_{jk} + \beta_t)_{q+\delta^s(j),\delta^v(k)} = CK(A_{\delta^s(j')\delta^v(k')} + \beta_t)_{q+j',k'}, \tag{17}$$

where $j = \delta^s(j')$ and $k = \delta^v(k')$. It is known that $U_K^V(A, \beta) = \prod_{(j,k)\in\mathcal{E}} CK(A_{jk} + \beta_t)_{q+j,k}$

$$SWAP'^T U_K^V(A, \beta_t)SWAP' = \prod_{(j,k)\in\mathcal{E}} SWAP'^T CK(A_{jk} + \beta_t)_{q+j,k}SWAP'$$

$$= \prod_{(j,k)\in\mathcal{E}} CK(A_{\delta^s(j')\delta^v(k')} + \beta_t)_{q+j',k'} = \prod_{(j',k')\in\mathcal{E}'} CK(A'_{\delta^s(j)\delta^v(k)} + \beta_t)_{q+j,k} \tag{18}$$

$$= U_K^V(A', \beta_t) = U_K^V(\Lambda_{\delta_s} A\Lambda_{\delta_v}, \beta_t),$$

where $\Lambda_{\delta_s}$ and $\Lambda_{\delta_v}$ are the permutation matrices of the permutation $\delta_s$ and $\delta_v$. The equation indicates the equivariance of the graph message interaction layer.

### D.2.2 PROOF OF THEOREM D.2

As mentioned earlier,

$$\Phi(A, H, \Theta) = \Phi(A, \mathbf{H}^V, \mathbf{H}^S, \Theta) = \{\langle0|U_{peqg}^\dagger(A, \mathbf{H}^V, \mathbf{H}^S, \Theta)O_i U_{peqg}(A, \mathbf{H}^V, \mathbf{H}^S, \Theta)|0\rangle\}_{i=1}^{q+p}, \tag{19}$$

and we have proven that $U_{peqg}$ is permutation equivariant. *i.e.*,

$$\widetilde{\Lambda}^\dagger \cdot U_{peqg}(A, \mathbf{H}^V, \mathbf{H}^S, \Theta) \cdot \widetilde{\Lambda} = U_{peqg}(\Lambda_p A\Lambda_q, \Lambda_q\mathbf{H}^V, \Lambda_p\mathbf{H}^S, \Theta). \tag{20}$$

Therefore, $\Phi(\Lambda_p A\Lambda_q, \Lambda_q\mathbf{H}^V, \Lambda_p\mathbf{H}^S, \Theta)$

$$= \{\langle0|U_{peqg}^\dagger(\Lambda_p A\Lambda_q, \Lambda_q\mathbf{H}^V, \Lambda_p\mathbf{H}^S, \Theta)O_i U_{peqg}(\Lambda_p A\Lambda_q, \Lambda_q\mathbf{H}^V, \Lambda_p\mathbf{H}^S, \Theta)|0\rangle\}_{i=1}^{q+p}$$

$$= \{\langle0|\widetilde{\Lambda}^\dagger U_{peqg}^\dagger(A, \mathbf{H}^V, \mathbf{H}^S, \Theta))\widetilde{\Lambda}O_i\widetilde{\Lambda}^\dagger U_{peqg}(A, \mathbf{H}^V, \mathbf{H}^S, \Theta))\widetilde{\Lambda}|0\rangle\}_{i=1}^{q+p}$$

$$= \{\langle0|\widetilde{\Lambda}^\dagger U_{peqg}^\dagger(A, \mathbf{H}^V, \mathbf{H}^S, \Theta))O_{\pi_{q+p}(i)} U_{peqg}(A, \mathbf{H}^V, \mathbf{H}^S, \Theta))\widetilde{\Lambda}|0\rangle\}_{i=1}^{q+p} \tag{21}$$

$$= \Lambda_{q+p}\Phi(A, \mathbf{H}^V, \mathbf{H}^S, \Theta),$$

which indicates $\Phi(A, H_V, H_S, \Theta)$ is a permutation equivariant mapping.

### D.2.3 PROOF OF COROLLARY D.1

**Corollary D.3.** *(From Equivariance to Invariance).* *If a mapping $\phi(A, H^V, H^S) \in \mathbb{R}^{q+p}$ is permutation equivariant, $\sum(\phi(A, H^V, H^S))$ is permutation invariant.*

*Proof.* Assume that $\phi(A, \mathrm{H}^V, \mathrm{H}^S) = [y_1, ..., y_{q+p}]^\top$, According to Definition 2, $\phi(\Lambda_p A \Lambda_q, \Lambda_q \mathrm{H}^V, \Lambda_p \mathrm{H}^S) = [y_{\pi_{q+p}(1)}, ..., y_{\pi_{q+p}(q+p)}]^\top$. Therefore, we can obtain $\sum(\phi(A, \mathrm{H}^V, \mathrm{H}^S)) = \sum_{i=1}^{q+p} y_i = \sum_{i=1}^{q+p} y_{\pi_{q+p}(i)} = \sum(\phi(\Lambda_p A \Lambda_q, \Lambda_q \mathrm{H}^V, \Lambda_p \mathrm{H}^S))$.

By Theorem D.2 and Corollary D.3, we can obtain $\phi_{\mathrm{fea}}(A, H) = \sum_{i=1}^{q+p} \Phi(A, H, \Theta_{fea})_i$ is permutation invariant. Similarly, the mapping of predicting objective value $\phi_{\mathrm{obj}}(A, H) = \sum_{i=1}^{q+p} \Phi(A, H, \Theta_{obj})_i$ also holds.

### D.2.4 AN INTUITIVE EXAMPLE OF GRAPH MESSAGE INTERACTION LAYERS

In graph message interaction layers, each edge is mapped into a two-qubit quantum gate acted on qubits representing two nodes. For example, here is a graph $G$ with three nodes $a, b, c$. After the first feature encoding layer, the quantum state $|\psi\rangle = |\psi_a\rangle \otimes |\psi_b\rangle \otimes |\psi_c\rangle$. If there is an edge connecting nodes $a$ and $b$, the model will apply a two-qubit quantum gate between qubit $q_a$ and qubit $q_b$. Suppose the used two-qubit gate is $R_{ZZ}(\theta) = \exp(-i\theta Z \otimes Z)$ gate, then it is equivalent to multiplying the quantum state by a matrix $U = R_{ZZ}(\theta) \otimes I$. Thus, the quantum state is changed as $|\psi'\rangle = U|\psi\rangle = (\exp(-i\theta Z \otimes Z) \otimes I)(|\psi_a\rangle \otimes |\psi_b\rangle \otimes |\psi_c\rangle) = (R_{ZZ}(\theta) \otimes I)(|\psi_{ab}\rangle \otimes |\psi_c\rangle) = |\psi'_{ab}\rangle \otimes |\psi_c\rangle$. In other words, the two-qubit gate alters the quantum states corresponding to nodes $a$ and $b$, thereby achieving the goal of information exchange.

## E GQGLA CAN DISTINGUISH MILP GRAPHS THAT GNN CANNOT DISTINGUISH

### E.1 WHY GNN FAILS ON THE GNN-INTRACTABLE MILPs

The discriminative power of GNN is defined as whether it can distinguish two non-isomorphic graphs. The representation power of GNN refers to its ability to approximate mappings with permutation equivariant/invariant properties. Moreover, Xu et al. (2018) have shown that GNNs are **at most** as powerful as the Weisfeiler-Lehman (WL) test Weisfeiler & Leman (1968) in distinguishing graph structures. The WL test is a well-known algorithm to identify whether two graphs are isomorphic or not, *i.e.*, whether two graphs are topologically identical. However, there are numerous WL test indistinguishable graphs, and the most well-known class is regular graphs, where every vertex has the same number of neighbors, *i.e.*, the same degree. According to the above, if a pair of graphs is indistinguishable by the WL test, GNN will also fail to distinguish them. In fact, the MILP graph dataset just contains numerous WL-indistinguishable graphs, so directly using GNN to represent general MILP graphs will lead to poor performance. Chen et al. (2023) extracted this subset of WL-indistinguishable graphs from the entire MILP dataset and named it the "GNN-intractable MILP". A variant of the WL test specially modified for MILP is provided in Algorithm 1. Based on the algorithm, it can intuitively show why the WL test cannot distinguish some non-isomorphic graphs, thereby showing why GNN cannot discriminate GNN-intractable MILPs.

---

**Algorithm 1** WL test for MILP-Graphs

---

**Input:** A graph instance $(A, H) \in \mathcal{G}_{q,p} \times \mathcal{H}_q^V \times \mathcal{H}_p^S$, and iteration limit $L > 0$.
Initialize with $c_v^0 = hash(f^V)$ for all $v \in V$, $c_s^0 = hash(f^S)$ for all $s \in S$.
**for** $l = 1$ **to** $L$ **do**
    $c_{v_i}^l = hash(c_{v_i}^{l-1}, \sum_{j=0}^{p-1} A_{i,j} hash(c_{s_j}^{l-1}))$, for all $v \in V$.
    $c_{s_j}^l = hash(c_{s_j}^{l-1}, \sum_{i=0}^{q-1} A_{i,j} hash(c_{v_i}^{l-1}))$, for all $s \in S$.
**end for**
**Output:** The multisets contain all colors $\{\{c_v^l : v \in V, c_s^l : s \in S\}\}$.

---

In the algorithm, $hash(\cdot)$ is a function that maps its input feature to a color in $\mathcal{C}$. The algorithm flow can be seen as follows. First, all nodes in $V \cup S$ are assigned to an initial color $c_v^0$ and $c_s^0$ according to their node features. Then, for each $v_i \in V$, the $hash$ function maps the previous color $c_{v_i}^{l-1}$ and aggregates the color of the neighbors of $\{c_{s_j}^{l-1}\}_{s_j \in \mathcal{N}(v_i)}$. Similarly, the $hash$ function maps

the previous color $c_{s_j}^{l-1}$ and aggregates the color of the neighbors of $\{c_{v_i}^{l-1}\}_{v_i \in \mathcal{N}(s_j)}$. This process is repeated until $L$ reaches the maximum iteration number. Finally, a histogram $h_G$ of the node colors can be obtained according to $\{\{c_v^l : v \in V, c_s^l : s \in S\}\}$, which can be used as a canonical graph representation. The notation $\{\{\cdot\}\}$ denotes a multiset, which is a generalisation of the concept of a set in which elements may appear multiple times. That is, the -WL test transforms a graph into a canonical representation. If the canonical representation of two graphs is equal, the WL test will consider them isomorphic.

Assume that there are two $n$-regular MILP graphs $G_1 = (A_1, H_1)$ and $G_2 = (A_2, H_2)$, where each node has $n$ neighbors in both graphs. And we set $A_1 \neq A_2$ and $H_1 = H_2$, *i.e.*, they have the same features but different topology structures (the connectivity of edges). For clarity, we set $\{f_i^V\}_{i=1}^q$ is the same, and $\{f_i^S\}_{j=1}^p$ is the same. Fig. 1 is an example of this type of graph. Initially, in the WL-test, $(c_v^0)_{g1} = (c_v^0)_{g2}$ and $(c_s^0)_{g1} = (c_s^0)_{g2}$. Then, the representation of each vertex is updated iteratively based on itself and information from its neighbors. In both graphs, each variable node has the same number of constraint nodes, and all constraint features are the same, so all $c_{v_i}^l$ are identical at this step. Similarly, each constraint node has the same number of variable nodes, so all $c_{s_j}^l$ are identical. Until the maximum number of iterations is reached, the algorithm will output the same representation for these two graphs. Therefore, the WL-test or GNNs cannot distinguish them.

### E.2 PROOF OF THE DISCRIMINATIVE POWER OF GQGLA BETTER THAN GNNs

**Theorem E.1.** *GQGLA can capture the difference in edge connectivity between two MILP graphs.*

*Proof.* Suppose that $G_1 = (A_1, H_1)$ and $G_2 = (A_2, H_2)$, and the two graphs have only different edge connectivity or edge weights, *i.e.*, $A_1 \neq A_2$. In GQGLA, the layer related to $A$ is the graph message interaction layer. As shown in Eq. 10, the sublayers related to $A$ are $U_K$ layers. All $U_K$ layers have similar structures. For clarity, we consider the following a simplified $U_K$ layer (where one qubit represents one node) to showcase how GQGLA captures the topological structure. *i.e.*,

$$U_K^V(A) = \exp(-\mathbf{i}(\sum_{(i,j) \in \mathcal{E}} A_{i,j}(I - \sigma_z)_i P_{q+j}^K)).$$

Note that $i \in \{1, ..., q\}$ and $j \in \{1, ..., p\}$, and $(I - \sigma_z)_i$ is equal to $\underbrace{I \otimes ... \otimes ...}_{i-1} \otimes (I - \sigma_z) \otimes \underbrace{... \otimes I}_{q-i} \otimes \underbrace{... \otimes I}_{p}$. Similarly, $P_{q+j}^K = \underbrace{I \otimes ... \otimes I}_{q} \otimes \underbrace{... \otimes}_{j-1} P^K \underbrace{\otimes ... \otimes I}_{p-j}$. Therefore, $(I - \sigma_z)_i P_{q+j}^K = \underbrace{I \otimes ... \otimes (I - \sigma_z)}_{i-1} \underbrace{\otimes ... \otimes I}_{q-i} \underbrace{\otimes ... \otimes}_{j-1} P^K \underbrace{\otimes ... \otimes I}_{p-j}$.

As we can see, for $(I - \sigma_z)_i P_{q+j}^K$, $I - \sigma_z$ appears in one of the first $q$ positions, and $P^K$ is acted in one of the last $p$ positions. For different edges, $(I - \sigma_z)_i P_{q+j}^K$ is different. More importantly, for different edge sets, $\sum_{(i,j) \in \mathcal{E}} A_{i,j}(I - \sigma_z)_i P_{q+j}^K$ is also different due to the properties of tensor products. Therefore, if $A_1 \neq A_2$, resulting in $U_K^V(A_1) \neq U_K^V(A_2)$, which indicates the GQGLA can capture the difference in edge connectivity between two MILP graphs. Although two non-isomorphic regular graphs have the same degree, the edge connectivity is different. That is, GQGLA can distinguish GNN-intractable MILP graphs that GNN fails to distinguish.

## F THE PERFORMANCE OF ALL POSSIBLE TYPES OF GQGLAS

Table 10 lists all possible selections of the quantum gate in GQGLA. The type represents $(P^w, P^D, P^K, P^G)$. For example, when the type is equal to $(\sigma_x, \sigma_z, \sigma_y, \sigma_z)$, it indicates $W(\theta) = R_X(\theta)$, $D(\theta) = R_Z(\theta)$, $CK(\theta) = CR_Y(\theta)$, and $G(\theta) = R_Z(\theta)$. To compare the performance of different settings, we test them on the task of predicting the feasibility of the GNN-tractable MILP dataset. In the experiment, we set the block of GQGLA is 8, the number of epochs to 15, the learning rate to 0.1, and the batch size to 16. Table 10 shows GQGLA with $(\sigma_x, \sigma_z, \sigma_y, \sigma_z)$ can achieve the best result, so we adopt this scheme in our all experiments.

Table 10: All possible selections of GQGLA and their performance results on predicting the feasibility of GNN-tractable MILP datasets. Results demonstrated that $(\sigma_x, \sigma_z, \sigma_y, \sigma_z)$ can achieve the best performance.

| Types | $(\sigma_x, \sigma_y, \sigma_z, \sigma_y)$ | $(\sigma_x, \sigma_z, \sigma_y, \sigma_z)$ | $(\sigma_y, \sigma_x, \sigma_z, \sigma_x)$ | $(\sigma_y, \sigma_z, \sigma_x, \sigma_z)$ | $(\sigma_z, \sigma_x, \sigma_y, \sigma_x)$ | $(\sigma_z, \sigma_y, \sigma_x, \sigma_y)$ |
|---|---|---|---|---|---|---|
| **Train** | 0.0598 | 0.0557 | 0.0724 | 0.0623 | 0.0635 | 0.0615 |
| **Test** | 0.0606 | 0.0574 | 0.0741 | 0.0625 | 0.0647 | 0.0635 |

## G  MILP DATASET GENERATION

For GNN-tractable MILPs, we first set the number of variables and constraints to $m$ and $n$.

- For each variable, $c_j \sim \mathcal{N}(0, 0.01)$, $l_j, u_j \sim \mathcal{U}(0, 2\pi)$. If $l_j > u_j$, then switch $l_j$ and $u_j$. The probability that $x_j$ is an integer variable is 0.5.
- For each constraint, $\circ_i \sim \mathcal{U}(\leq, =, \geq)$ and $b_i \sim \mathcal{U}(-1, 1)$.
- After randomly generating all the MILP samples, we use the WL test algorithm to calculate their graph representation for each instance, ensuring that there are no duplicate graph representations in the dataset, so that we can determine that this dataset does not contain WL-test indistinguishable pairs of MILP instances.

The GNN-intractable dataset is constructed by many pairs of WL indistinguishable graphs, and Fig. 1 in our paper is a GNN-intractable example, which is a pair of non-isomorphic graphs that cannot be distinguished by the WL-test or by GNNs. The GNN-intractable dataset randomly generates 2000 GNN-intractable MILPs with 12 variables and 6 constraints, and there are 1000 feasible MILPs with feasible optimal solutions while the others are infeasible. We construct the $(2k-1)$-th and $2k$-th problems via the following approach, where $(1 \leq k \leq 500)$.

- Sample $J = \{j_1, j_2, ..., j_6\}$ as a random subset of $\{1, 2, ..., 12\}$ with 6 elements. **1)** For $j \in J$, $x_j \in \{0, 1\}$, i.e., $x_j$ is a binary integer variable. **2)** For $j \notin J$, $x_j$ is a continuous variable with bounds $l_j \sim \mathcal{U}(0, \pi)$, $u_j \sim \mathcal{U}(0, \pi)$. If $l_j > u_j$, then switch $l_j$ and $u_j$.
- $c_1 = ... = c_{12} = 0.01$.
- The constraints for the $(2k-1)$-th problem (feasible) is $x_{j_1} + x_{j_2} = 1$, $x_{j_2} + x_{j_3} = 1$, $x_{j_3} + x_{j_4} = 1$, $x_{j_4} + x_{j_5} = 1$, $x_{j_5} + x_{j_6} = 1$, $x_{j_6} + x_{j_1} = 1$. For example, $x = (0, 1, 0, 1, 0, 1)$ is a feasible solution.
- The constraints for the $2k$-th problem (infeasible) is $x_{j_1} + x_{j_2} = 1$, $x_{j_2} + x_{j_3} = 1$, $x_{j_3} + x_{j_1} = 1$, $x_{j_4} + x_{j_5} = 1$, $x_{j_5} + x_{j_6} = 1$, $x_{j_6} + x_{j_4} = 1$.

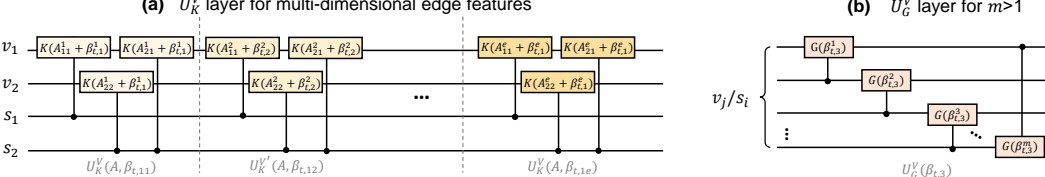

Figure 9: (a) The controlled $K$ layer for multi-dimensional edge features. (b) The controlled $G$ layer for $m > 1$, where $m$ is the number of qubits representing one node $v_j$ or $s_i$.

## H  THE CAPACITY OF GQGLA

The proposed framework can also be used for situations where the edge dimension is more than one. As shown in Fig. 9 (a), the controlled $K$ gates can be repeatedly applied with different edge dimensions. In addition, Fig. 9 (b) illustrates the detailed circuit when the number of qubits representing one node is more than one. Moreover, although GQGLA is designed for MILP graphs, i.e., bipartite graphs, it can be easily extended to arbitrary graphs. Specifically, in the quantum graph message interaction layer, we can apply symmetric two-qubit gates, such as $R_{ZZ}$ gate, between two nodes, representing edges in an arbitrary graph and preserving the edge permutation invariance. The remaining structure can remain completely unchanged when applied to an arbitrary graph.

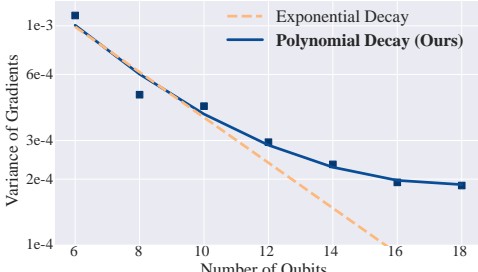

Figure 10: Variance v.s qubits (log-linear scale).

It has been shown that general QNNs may suffer from barren plateau (McClean et al., 2018), *i.e.*, the loss gradients vanish exponentially with the system size, causing the trainability of QNNs becomes an important issue. However, Fig. 10 shows the variance of our cost function partial derivatives for a parameter in the middle of GQGLA. The variance only decreases polynomially with the system size, which shows that GQGLA has good trainability.

## I   THE DETAILS OF $U_K$ LAYER

In the training process, the types (e.g., $RX(\theta)$, $RY(\theta)$, or $RZ(\theta)$) and positions of gates are fixed, but their internal parameters are not fixed in the quantum machine learning. Therefore, it should be noted that when implementing the proposed graph message interaction layer, there is actually a controlled $K$ gate between each variable node and each constraint node (like a fully connected bipartite graph). However, if there is no edge between the two nodes, we will set the parameters of the corresponding controlled K gate to 0. $CK(0)$ is an identity matrix that will not affect the overall circuit. Specifically, in code engineering, we first obtain the binary adjacency matrix $T$ of the bipartite graph. $T_{ij} = 1$ indicates that there is an edge connecting constraint node $s_i$ and variable node $v_j$, while $T_{ij} = 0$ indicates that there is no edge connecting them. Then, $CK(T_{ij}(A_{ij} + \beta))$ is acted on the two qubits representing constraint node $s_i$ and variable node $v_j$. As we can see, when there is an edge between the two qubits, $CK(A_{ij} + \beta)$ is acted on the two qubits, while when there is no edge between the two qubits, $CK(0)$ is acted on the two qubits. In this way, we can achieve the modeling of the graph message interaction layer under the premise of fixed gate types and positions.

## J   THE IMPLEMENTATION ON REAL QUANTUM DEVICES

Our proposed GQGLA quantum circuit consists of only simple single-qubit and two-qubit gates, making it easy to deploy on existing NISQ devices. To verify this, we use the Qiskit package and the IBM Quantum platform to directly execute our circuits on real IBM quantum hardware. The circuits are first compiled and optimized, then mapped to the real quantum hardware's topology using the $generate\_preset\_pass\_manager$ function. Subsequently, the expectation values of the quantum circuits are estimated using the *Estimator V2* primitive. The Estimator primitive supports three resilience levels: Resilience Level 0 represents no error mitigation techniques are applied. Resilience Level 1 applies readout error mitigation and measurement twirling using a model-free technique known as Twirled Readout Error eXtinction (TREX) (Van Den Berg et al., 2022). Resilience Level 2 includes the error mitigation techniques from Level 1 and further applies gate twirling and the Zero Noise Extrapolation (ZNE) (Temme et al., 2017).

We conducted experiments using 10 minutes of free usage on the 127-qubit *IBM Brisbane* quantum computer, and evaluate the mean square error (MSE) compared to ground truth on MILP dataset with three different scales, as shown in the table below. The results demonstrate that employing error mitigation strategies can improve performance. Higher resilience levels produce more accurate results, but at the cost of increased processing time, which is a trade-off between cost and accuracy.

|  | 18 qubit | 26 qubit | 36 qubit |
|---|---|---|---|
| Noise (ibm_brisbane) | 0.2627 | 0.2850 | 0.3461 |
| resilience_level = 1 | 0.2107 | 0.2428 | 0.3336 |
| resilience_level = 2 | 0.2077 | 0.2174 | 0.3176 |

Table 11: Performance comparison across different qubit numbers and resilience levels.

In addition, we studied the parameter transferability and the impact of noise on performance. We train on the 20-qubit GNN-intractable MILPs and transfer their parameters into problems with 30, 40, and 50 qubits to test. $n$ qubit represents a MILP graph dataset containing 200 different graphs with $n$ nodes.

Specifically, we use Qiskit's (Javadi-Abhari et al., 2024) matrix product state simulator, as well as IBM's noise model FakeWashingtonV2, with the backends built to mimic the behaviors of IBM Quantum systems. As shown in Table 12, the model parameters trained on the 20-qubit remain effective for the 50-qubit data, benefiting from GQGLA's parameter-sharing mechanism. This means that the parameters

Table 12: Parameter transferability across qubits under noiseless and noisy conditions.

| Qubit | 20 (base) | 30 | 40 | 50 |
|---|---|---|---|---|
| Noiseless | 1.0 | 0.967 | 0.908 | 0.862 |
| Nosiy | 0.995 | 0.928 | 0.864 | 0.825 |

trained on small qubit datasets can provide a good initialization for larger qubit datasets, speeding up the training. Moreover, while noise does have an impact on performance, the effect is not big.

