# OpenReview forum: "On Designing General and Expressive Quantum Graph Neural Networks with Applications to MILP Instance Representation"
_ICLR.cc/2025/Conference — ICLR 2025 Poster_

### Official Review · Reviewer_9Vep · 2024-11-04

**Soundness:** 2
**Presentation:** 3
**Contribution:** 3
**Rating:** 5
**Confidence:** 2

**Summary:**

The authors proposed a quantum machine learning method to solve mixed integer programming via graph learning.

**Strengths:**

The proposed method is able to encode MILP graphs that are known to be intractable by message passing GNNs. The authors show that this method is able to surpass GNN baselines, especially on GNN-intractable problems.

**Weaknesses:**

The examples of GNN-intractable MILP problems are specific to message passing GNNs. Various works on higher-order GNNs have been proposed to overcome this issue in classical GNN architectures. Indeed, the authors provided experimental comparison to graph transformers; but basing the main motivation on a weak GNN lacks persuasiveness.

**Questions:**

Is the proposed method still superior comparing to higher order GNNs? Examples include subgraph GNNs that can distinguish the graphs in figure 1.

---

> ### Author Response · Authors · 2024-11-19
> **Response to Reviewer 9Vep (Part 1)**
>
> Thanks for your review and feedback. Below is our detailed response.
>
> > **W1 & Q1: The examples of GNN-intractable MILP problems are specific to message passing GNNs.  Indeed, the authors provided experimental comparison to graph transformers; but basing the main motivation on a weak GNN lacks persuasiveness. Is the proposed method still superior comparing to higher order GNNs? Examples include subgraph GNNs that can distinguish the graphs in figure 1.**
>
> In Table 8 of the previous PDF, we compared our method with higher-order GNNs including the subgraph method NGNN [2], $\delta$-k-LGNN [5], Graphormer [8], GSN [9] on the difficult graph datasets, but it seems it was not presented clearly enough. Therefore, we have rewritten this part and added more advanced GNNs to our [updated pdf](https://openreview.net/pdf?id=IQi8JOqLuv), see Section 5.4 highlighted in blue in the PDF. In addition, we have provided another experiments with more advanced GNNs in Table 9 of our [updated pdf](https://openreview.net/pdf?id=IQi8JOqLuv). Next, we provide the details of these two experiments.
>
> 1. To verify the expressiveness of our methods, we conducted an experiment on the recent benchmark **BREC** [1] used to evaluate GNN expressiveness, compared higher order GNNs, including
>
> *  **Subgraph-based methods:** NGNN [2], DS-GNN [3] and KP-GNN [4]
> *  **K-WL GNNs:** $\delta$-k-LGNN [5] and PPGN [6]
> *  **Random GNN:** DropGNN [7]
> *  **Transformer-based model:** Graphormer [8]
> *  **Substructure-based model:** GSN [9].
>
> | Methods   | NGNN [2] $\quad$ | DS-GNN [3] $\quad$| KP-GNN [4] $\quad$| $\delta$-k-LGNN [5] $\quad$ | PPGN [6] $\quad$| DropGNN [7] $\quad$| Graphormer [8] $\quad$ | GSN [9] $\quad$ | VQGLA (Ours) $\quad$ |
> | --------------------- | ---------------- | ------ | ------ | --------------------------- | ---- | ------- | ---------------------- | --------------- | ----- |
> | **Basic**             | 0.983            | 0.967  |   1.0     | 1.0                      |  1.0     |   0.867  | 0.267                  | 1.0             | 1.0   |
> | **Regular**           | 0.343            | 0.343  |   0.757   | 0.357                    |  0.357  |    0.293  | 0.086                  | 0.707           | 0.964 |
> | **Extension** $\quad$ | 0.59             | 1.0    |   0.98    | 1.0                      |  1.0    |    0.820  | 0.41                   | 0.95            | 1.0   |
>
> The *Basic* dataset in BREC consists of 1-WL-indistinguishable graphs and is designed to be non-regular. The *Regular* dataset contains strongly regular graphs, where 1-WL and 3-WL tests fail, and includes 4-vertex condition graphs and distance-regular graphs, further increasing the dataset’s complexity. The *Extension* graphs bridge the gap between 1-WL and 3-WL, offering a more granular comparison for evaluating models beyond 1-WL. As shown in the table, the proposed VQGLA achieves not only good performance on the *Basic* and *Extension* datasets but also outperforms other higher-order GNNs on the challenging *Regular* dataset. This is attributed to the ability of the quantum circuit we designed to leverage the vast Hilbert space to capture the intricate topological structure of the graph. Further details are provided in Appendix E.2 of our paper.
>
> 2. We also verify our performance on three commonly used graph classification datasets, where PROTEINS contains graphs with up to 620 nodes. We leverage the graph sampling technique [10] to expand the scale of problems that our approach can handle. The accuracy of our S-VQGLA and different advanced GNNs are compared, including message-passing-based GNNs GraphSAGE [11], GIN [12], and GAT [13], and **subgraph-based methods** PPGN [6], QS-CNN [14] and Deep WL SGN [15], and **transformer-based methods** U2GNN [16] and SEG-BERT [17].
>
> | Methods                      | PROTEINS $\quad$ | MUTAG  $\quad$  $\quad$| PTC   $\quad$ |
> | ---------------------------- | ---------------- | -------------- | ------------- |
> | GraphSAGE [11]                | 75.9 ± 3.2       | 85.1 ± 7.6     | 63.9 ± 7.7    |
> | GIN  [12]                    | 76.20 ± 2.8      | 89.40 ± 5.6    | 64.6 ± 7.0    |
> | GAT  [13]                       | 74.70 ± 2.2      | 89.40 ± 6.1    | 66.70 ± 5.1   |
> |                              |                  |                |               |
> | PPGN [6]                        | 76.66 ± 5.6      | 89.44 ± 8.1    | 62.94 ± 6.6   |
> | QS-CNN [14]                     | 78.2 ± 4.6       | 93.1 ± 4.7     | 66.0 ± 4.4    |
> | Deep WL SGN  [15]             | 76.78 ± 2.4      | 93.68 ± 5.1 | 65.88 ± 5.1   |
> |                              |                  |                |               |
> | U2GNN  [16]                   | 78.53 ± 4.1      | 89.97 ± 3.6    | 69.63 ± 3.6   |
> | SEG-BERT [17]                     | 77.20 ± 3.1      | 90.85 ± 6.6    | 68.86 ± 4.2   |
> |                              |                  |                |               |
> | **S-VQGLA  (Ours)**  $\quad$ | 80.40 ± 2.6  | 91.64 ± 3.1   | 68.57 ± 2.9    |

---

> ### Author Response · Authors · 2024-11-19
> **Response to Reviewer 9Vep (Part 2)**
>
> As shown in the above table, our method can achieve the best performance on the PROTEINS, and slightly worse than the Deep WL SGN and U2GNN on the MUTAG and PTC, respectively. The experiment results show that VQGLA can still achieve good performance on larger-scale graph datasets.
>
> In the end, we would like to mention the reason why, apart from Table 8 and Tbale 9, the paper chooses message-passing GNNs as a comparison.
>
> **First**, quantum machine learning is still in its infancy (e.g., the current field is still struggling with the design of the quantum counterpart of Transformer), especially quantum graph machine learning. Existing quantum GNNs are either quantum-classical hybrid methods that cannot be effectively applied to current NISQ devices, or cannot handle edge features, or fail to satisfy the permutation equivariance property of graph nodes. To address these limitations, we proposed versatile quantum graph learning architecture (VQGLA), which can be regarded as *a basic backbone* of graph learning *in the field of quantum computing*. VQGLA chooses message-passing GNNs as the primary comparison because they are the most fundamental and widely used method in classical graph learning. We aim for VQGLA to play a similar foundational role in quantum graph learning as message-passing GNNs do in classical graph learning.
>
> **Second**, VQGLA only utilizes first-order graph information and does not take higher-order graph structures into account, as higher-order GNNs do. As a foundational backbone, VQGLA can also be extended into a more powerful version by incorporating higher-order graph structures. We have added the above explanation to our [updated pdf](https://openreview.net/pdf?id=IQi8JOqLuv), see Section 2 highlighted in blue.
>
> ---
> We hope that our responses and modifications have addressed your concerns. If you have any other concerns, we are more than glad to provide further responses. We look forward to receiving your feedback soon.
>
> **References**:
>
> [1] *An empirical study of realized gnn expressiveness, ICML 2024.*
>
> [2] *Nested graph neural networks, NeurIPS 2021.*
>
> [3] *Equivariant subgraph aggregation networks, ICLR 2022.*
>
> [4] *How powerful are k-hop message passing graph neural networks, ICLR 2022.*
>
> [5] *Weisfeiler and leman go sparse: Towards scalable higher-order graph embeddings, NeurIPS 2020.*
>
> [6] *Provably powerful graph networks, NeurIPS 2019.*
>
> [7] *Dropgnn: Random dropouts increase the expressiveness of graph neural networks, NeurIPS 2021.*
>
> [8] *Do transformers really perform badly for graph representation?  NeurIPS 2021.*
>
> [9] *Improving graph neural network expressivity via subgraph isomorphism counting, TPAMI 2022.*
>
> [10] *GraphSAINT: Graph sampling based inductive learning method, ICLR 2020.*
>
> [11] *Inductive representation learning on large graphs. NeurIPS 2017.*
>
> [12] *How Powerful are Graph Neural Networks? ICLR 2019.*
>
> [13] *Graph attention networks, ICLR 2018.*
>
> [14] *Quantum-based subgraph convolutional neural networks, PR 2019.*
>
> [15] *Subgraph networks with application to structural feature space expansion. TKDE 2019.*
>
> [16] *Universal Graph Transformer Self-Attention Networks, WWW 2022.*
>
> [17] *Segmented graph-bert for graph instance modeling, arXiv 2020.*

---

> ### Author Response · Authors · 2024-11-23
> **Looking forward to hearing your feedback**
>
> Dear Reviewer 9Vep:
>
> As the discussion phase of ICLR draws to a close, we would like to kindly follow up to ensure that our responses have addressed your concerns. Specifically, in our previous response, we provided comparisons with higher-order GNN models on hard graph datasets, highlighting the superior expressiveness of our approach.
>
> If you have any remaining questions or concerns, we would be delighted to provide further details or clarifications.
>
> We greatly appreciate your time and feedback and look forward to hearing your thoughts.
>
> Best regards,

---

> ### Comment · Area_Chair_UStG · 2024-11-25
> **ICLR Public Discussion Phase Ending Soon**
>
> Dear Reviewer,
>
> This is a kind reminder that the discussion phase will be ending soon on November 26th. Please read the author responses and engage in a constructive discussion with the authors.
>
> Thank you for your time and cooperation.
>
> Best,
>
> Area Chair

---

> ### Author Response · Authors · 2024-11-27
> **Looking forward to hearing your feedback**
>
> Dear Reviewer 9Vep:
>
> I hope this message finds you well. We have updated the previous response, which added more comparison methods and datasets to the two experiments. As the rebuttal phase deadline is approaching, we would like to kindly request your further feedback to ensure a comprehensive revision and to ensure our clarifications have addressed your concerns effectively. Your feedback is invaluable, and we greatly appreciate your time and insights.
>
> Best regards,

---

> ### Author Response · Authors · 2024-11-30
> **Looking forward to your feedback**
>
> Dear Reviewer 9Vep:
>
> We hope that our responses could adequately address your concerns. As the deadline for this discussion phase approaches, we warmly welcome further discussion regrading any additional concerns that you may have.
>
> Thank you for the time and appreciation that you have dedicated to our work.
>
> Best regards,
>
> Authors of submission 2442

---

### Official Review · Reviewer_x2kb · 2024-11-08

**Soundness:** 3
**Presentation:** 4
**Contribution:** 3
**Rating:** 8
**Confidence:** 4

**Summary:**

The authors present a versatile quantum graph learning architecture (VQGLA) method for solving GNN-intractable mixed-integer linear programming (MILP). The VQGLA encodes features of nodes and edges in quantum circuits, which maintain node permutation equivariance. The new method combine the power of quantum computing and GNN, exhibiting superior discriminative power over classical GNNs.

**Strengths:**

The authors present a well established method for MILPs. By introducing quantum computing, this new method is able to solve GNN-intractable MILPs. Numerical results can fully support this conclusion. I believe that this work exhibits an interesting application of quantum computing for solving real-world optimization problems.

**Weaknesses:**

The quantum circuits that are used to encode the constraint node are not straightforward. Could the authors explain how to map mathematical operators onto quantum circuits?
This work exhibits excellent performance of VQGLA in solving some MILPs. It is interesting to discuss the scalability and quantum circuit complexity of this method for general graph structures.
Since quantum neural network is an interdiscipline, the authors would better clarify how to turn a real-world problem into a quantum computing problem.

**Questions:**

The authors show a good parameter transferability. I wonder if these graphs have the same structures?
The authors compare the VQGLA and the HEA in their numerical results. Did they encode the graph information in the HEA?

---

> ### Author Response · Authors · 2024-11-19
> **Response to Reviewer x2kb (Part 1)**
>
> Thank you for acknowledging our work, and your detailed feedback and suggestions have been immensely helpful. Below is our detailed response.
>
> > **W1: The quantum circuits that are used to encode the constraint node are not straightforward. Could the authors explain how to map mathematical operators onto quantum circuits?**
>
> Assume there is a constraint $A_{11} x_1 + A_{12} x_2 \circ b$, where $\circ \in \\{\leq, =, \geq \\}$. In MILP graph, the according constraint node has a feature vector $[b, \circ]$, where $\circ \in \\{0,1,2\\}$. Consistent with [1], the mathematical operators $\\{\leq, =, \geq\\}$ are mapped into numerical value $\\{0,1,2\\}$, respectively. Then, we encode the value of the node feature into the quantum gate by angle encoding. Specifically, as shown in [this figure](https://postimg.cc/64hbpgpL), the feature value is used as the parameter of quantum gate $D \in \\{R_x(\theta), R_y(\theta), R_z(\theta)\\}$ in the feature encoding layer.  It seems that our paper did not mention mapping mathematical operators to numerical values, so we have added the above explanation to [our updated PDF](https://openreview.net/pdf?id=IQi8JOqLuv). Thank you for pointing this out.
>
> *[1] On representing mixed-integer linear programs by graph neural networks, ICLR 2023.*
>
>
> > **W2. This work exhibits excellent performance of VQGLA in solving some MILPs. It is interesting to discuss the **scalability and quantum circuit complexity** of this method for general graph structures. Since quantum neural network is an interdiscipline, the authors would better clarify how to turn a **real-world problem** into a quantum computing problem.**
>
> Thanks for your suggestions. For general graph structures with $N$ nodes and $E$ edges, the node feature encoding layer involves $O(N)$ single-qubit gates, while the graph message interaction layer introduces $O(E)$ two-qubit gates. For a circuit with $T$ blocks, the total number of gates scales as $O(T(N+E))$. Typically, $E > N > T$, so the number of quantum gates can be approximately considered to have a linear complexity with respect to the number of edges, i.e., $O(E)$. In addition, a graph with $N$ nodes typically requires $O(N)$ qubits.
>
> It is worth mentioning that our designed model can be easily deployed on current quantum hardware. On the one hand, different from models [2,3] based on quantum linear algebra that require complex quantum gates that are hard to compile, our VQGLA only contains the simple single-qubit quantum gate and two-qubit gate. On the other hand, different from models [4,5] based on hybrid quantum-classical layers that require very time-consuming amplitude encoding as well as tomography readout of quantum states, our VQGLA only uses lightweight angle encoding and Pauli-z measurement readout.
>
> However, in the current NISQ (Noisy Intermediate-Scale Quantum computing) era, both real quantum devices and classical simulators have a limited number of qubits. While our VQGLA is designed with the expectation of the arrival of Fault-Tolerant Quantum Computing, we also expect its effectiveness in the NISQ era. Therefore, inspired by how classical GNN algorithms handle large graph computations with limited resources, we have presented S-VQGLA to process large graphs with limited qubits. Specifically, we first employ the graph sampling technique GraphSAINT [6] to extract appropriately connected subgraphs, then apply our VQGLA model to these sample subgraphs and combine the obtained information of these subgraphs together so that the training process overall learns information of the full graph. In this way, for a graph with $N$ nodes, we can sample $m$ subgraphs for training, with each subgraph using at most $k$ qubits. This training approach allows us to reduce the qubit number from $O(N)$ to $O(m)$, which can be regarded as resource requirements are reduced from $O(2^N)$ to $O(m \cdot 2^k)$, where $m,k<N$.
>
> Here we provide an experiment to verify the performance of VQGLA on a commonly used dataset, PROTEINS, which contains proteins that are classified as enzymes or non-enzymes. This dataset consists of 1,113 graphs, and each graph has between 4 to **620** nodes. We extract appropriately connected subgraphs with at most 14 nodes, which can be processed using our 14-qubit VQGLA with 6 blocks. All subgraphs are input as a batch into the VQGLA, and we combine their results to predict the classification of the full graph. The following table shows the graph classification results for our S-VQGLA and other classical GNN algorithms.

---

> > ### Comment · Reviewer_x2kb · 2024-11-25
> >
> > The authors have satisfactorily addressed most of my concerns

---

> > > ### Author Response · Authors · 2024-11-25
> > > **To Reviewer x2kb**
> > >
> > > Dear Reviewer x2kb,
> > >
> > > We are truly pleased that our response addressed your concerns and questions. Thank you for the time and effort you dedicated to reviewing our work.
> > >
> > > With gratitude,

---

> ### Author Response · Authors · 2024-11-19
> **Response to Reviewer x2kb (Part 2)**
>
> | Methods      | Accuracy (%) |
> | ---------------- | --------------- |
> | GraphSAGE [7]      | 73.0          |
> | GIN  [8]            | 76.2          |
> | GAT-GC [9]        | 76.8          |
> | 2-WL-GNN  [10]      | 76.5          |
> | U2GNN  [11]         | 78.5          |
> | UGT  [12]            | 80.1          |
> | **S-VQGLA  (Ours)**  $\quad$  | **80.4**          |
>
>
> As we have observed, S-VQGLA not only outperforms message-passing-based GNN methods [7,8,9], but also slightly exceeds the recently proposed Graph Transformer-based methods [11,12]. The graph sampling technique enables our approach to be effectively applied to larger graphs under limited qubit resources, while it also can save qubit resources for future computations. We have added the above description and experiment to our updated PDF, see Section 6 highlighted in blue in the PDF.
>
> **References**
>
> [2] Quantum support vector machine for big data classification, Physical review letters, 2014.
>
> [3] Quantum linear algebra is all you need for Transformer architectures, 2024.
>
> [4] Learning graph convolutional networks based on quantum vertex
> information propagation, TKDE 2021.
>
> [5] Hybrid quantum-classical graph convolutional network, 2021.
>
> [6] GraphSAINT: Graph sampling based inductive learning method, ICLR 2020.
>
> [7] Inductive representation learning on large graphs. NeurIPS 2017.
>
> [8] How Powerful are Graph Neural Networks? ICLR 2019.
>
> [9] Improving Attention Mechanism in Graph Neural Networks via Cardinality Preservation, IJCAI 2020.
>
> [10] A Novel Higher-order Weisfeiler-Lehman Graph Convolution, ACML, 2020.
>
> [11] Universal Graph Transformer Self-Attention Networks, WWW 2022.
>
> [12] Transitivity-Preserving Graph Representation Learning for Bridging Local Connectivity and Role-Based Similarity, AAAI 2024.
>
>
> > **Q1. The authors show a good parameter transferability. I wonder if these graphs have the same structures?**
>
> In the table of parameter transferability, the result of qubit $n$ is obtained by experimenting on a MILP graph dataset containing 200 different graphs with $n$ nodes. We report the result of four datasets with 20, 30, 40, and 50 nodes, and they have different graph structures. We have added the explanation to our updated PDF. Thanks for your suggestion, which makes our article clearer.
>
>
> > **Q2. The authors compare the VQGLA and the HEA in their numerical results. Did they encode the graph information in the HEA?**
>
> HEA is a problem-agnostic quantum model, and its circuit structure is typically fixed, making it unable to encode edge information of a graph. However, we use it to encode the node feature information of the graph. We aim to use it as a problem-agnostic baseline to show the necessity of problem-specific models for graph problems. We have added the above explanation to our updated PDF. Thanks for your suggestion, which makes our article clearer.
>
> ---
> We hope that our responses and modifications can address your concerns. Thank you very much for your suggestion, which has improved the quality of our work.

---

> > ### Comment · Reviewer_x2kb · 2024-11-25
> >
> > The authors have provided reasonable responses to my comments.

---

> ### Comment · Area_Chair_UStG · 2024-11-25
> **ICLR Public Discussion Phase Ending Soon**
>
> Dear Reviewer,
>
> This is a kind reminder that the discussion phase will be ending soon on November 26th. Please read the author responses and engage in a constructive discussion with the authors.
>
> Thank you for your time and cooperation.
>
> Best,
>
> Area Chair

---

### Official Review · Reviewer_xDyW · 2024-11-09

**Soundness:** 3
**Presentation:** 3
**Contribution:** 3
**Rating:** 6
**Confidence:** 4

**Summary:**

This paper proposes a new quantum-driven graph learning model called VQGLA, which aims to overcome the limitations of classical Graph Neural Networks (GNNs) in representing mixed-integer linear programming (MILP) problems. VQGLA leverages quantum computing to provide superior discriminative capabilities for complex optimization tasks like MILP representation. The proposed model introduces a node feature layer, a graph message interaction layer, and an auxiliary layer, ensuring permutation equivariance and flexible circuit configurations. Experimental results demonstrate VQGLA's effectiveness compared to classical GNNs on both GNN-tractable and GNN-intractable datasets.

**Strengths:**

1. The paper clearly articulates the limitations of classical GNNs in representing MILP problems and makes a strong case for the introduction of a quantum-driven solution. The use of quantum machine learning (QML) is well-motivated, especially in scenarios where classical methods are fundamentally limited, such as GNN-intractable graph representations.

2. The design of VQGLA, including the node feature layer, graph message interaction layer, and auxiliary layer, is well-thought-out and comprehensive. The incorporation of quantum entanglement to enhance discriminative power is innovative, and the theoretical proof of permutation equivariance strengthens the rigor of the work.

3. The experimental results are compelling. VQGLA's superior performance over classical GNNs is demonstrated through multiple benchmarks, including both GNN-tractable and GNN-intractable MILP datasets. The comparison to other quantum and classical methods is thorough, and the reported results support the claims made in the paper.

4. The paper demonstrates the versatility of VQGLA by applying it to various graph tasks, including graph classification, graph regression, and node property prediction. The adaptability of the model to different levels of graph tasks, as well as its ability to handle edge features, is a significant advantage.

**Weaknesses:**

1. While the paper emphasizes the benefits of using quantum methods, it lacks a detailed discussion on the computational complexity and scalability of VQGLA. The use of fully quantum circuits for large graph instances might introduce significant resource requirements, and this limitation is not addressed sufficiently. It would be helpful if the authors provided more insights into the scalability of VQGLA with respect to graph size and the number of quantum gates required.

2. The explanation of the quantum operations, especially in the auxiliary and graph message interaction layers, could be clearer. The use of quantum gates and entanglement is discussed at a high level, but more detailed explanations and diagrams would make it easier for readers without extensive quantum computing backgrounds to understand the exact role of each component in the quantum circuit.

3. Although VQGLA is compared to classical GNNs and a few quantum models, it would be beneficial to include more comparisons with state-of-the-art classical GNN architectures that have shown effectiveness in handling complex graph tasks, such as Graph Transformers or other hybrid models. This would provide a more comprehensive understanding of how VQGLA stands relative to the best classical methods.

4. The implementation of VQGLA on near-term quantum devices is briefly mentioned, but the impact of quantum noise on the model's performance is not explored in depth. Quantum hardware is prone to noise, and the absence of a discussion on error mitigation strategies or the robustness of VQGLA under noisy conditions weakens the paper's practical relevance.

**Questions:**

1. Can the authors provide more insights into the scalability of VQGLA, especially concerning the number of quantum gates required for larger graph instances?
2. How does the presence of quantum noise impact the performance of VQGLA? Are there any error mitigation strategies that could be employed?
3. Could the authors include a comparison with more advanced classical GNN models, such as Graph Transformers, to highlight the specific advantages of the quantum-driven approach?
4. Is there any practical guidance on implementing VQGLA on near-term quantum hardware? A discussion on the hardware requirements would be very helpful for future work.

---

> ### Author Response · Authors · 2024-11-19
> **Response to Reviewer xDyW (Part 1)**
>
> Thanks for your review and feedback, and we deeply appreciate your time and effort in reviewing our paper. Below is our detailed response.
>
> > **W1&Q1: While the paper emphasizes the benefits of using quantum methods, it lacks a detailed discussion on the computational complexity and scalability of VQGLA. The use of fully quantum circuits for large graph instances might introduce significant resource requirements, and this limitation is not addressed sufficiently. It would be helpful if the authors provided more insights into the scalability of VQGLA with respect to graph size and the number of quantum gates required. Can the authors provide more insights into the scalability of VQGLA, especially concerning the number of quantum gates required for larger graph instances?**
>
> Thanks for your suggestions. Next, we will provide an analysis of the complexity and scalability of our circuit. Moreover, to address the limitations of quantum resources in the current NISQ (Noisy Intermediate-Scale Quantum) era, we present S-VQGLA, which uses a graph sampling strategy to apply our method to large-scale graph instances.
>
> - **The complexity of our VQGLA**
>
>     - For a graph with $N$ nodes and $E$ edges, the node feature encoding layer involves $O(N)$ single-qubit gates, while the graph message interaction layer introduces $O(E)$ two-qubit gates. For a circuit with $T$ blocks, the total number of gates scales as $O(T(N+E))$. Typically, $E > N > T$, so the number of quantum gates can be approximately considered to have a linear complexity with respect to the number of edges, i.e., $O(E)$.
>
>     - Note that the overall time complexity of our model is low and can be readily deployed on current quantum hardware. On the one hand, different from models [2,3] based on quantum linear algebra that require numerous multi-qubit quantum gates that are hard to compile, our VQGLA only contains the simple single-qubit quantum gate and two-qubit gate. On the other hand, different from models [4,5] based on hybrid quantum-classical layers that require very time-consuming amplitude encoding as well as tomography readout of quantum states, our VQGLA only uses lightweight angle encoding and Pauli-Z measurement readout.
>
> - **The scalability of our VQGLA and the presented S-VQGLA**
>
>     - For the scalability of VQGLA, the number of required qubits is more critical than the number of gates, because qubits require exponential resources while the cost of adding gates at a fixed qubit is small. For a graph with $N$ nodes, our VQGLA typically requires $O(N)$ qubits. However, in the current NISQ era, both real quantum devices and classical simulators have a limited number of qubits. Classical simulations can usually simulate up to 50 qubits, and current available largest quantum hardware is 127-qubit IBM quantum computer. While our VQGLA is designed with the expectation of the arrival of Fault-Tolerant Quantum Computing (with enough qubits), we also expect its effectiveness in the NISQ era. Therefore, inspired by how classical GNN algorithms handle large graph computations with limited resources, we have presented S-VQGLA to process large graphs with limited qubits. Specifically,
> 1. We employ the graph sampling technique GraphSAINT [1] to extract appropriately connected subgraphs so that little information is lost when propagating within the subgraphs. Here, we use the edge sampler from GraphSAINT. For each edge $(u, v)$, calculate its sampling probability $P((u, v))$, which is proportional to $(1/deg(u)) + (1/deg(v))$. Intuitively, if two nodes $u$ and $v$ are connected to each other and have few neighbors, then they have a greater influence on each other, so the sampling probability of this edge should be higher. According to the sampling probability $P((u, v))$, randomly select a predetermined number of edges to generate subgraphs. We can control the sizes of the sample subgraphs to ensure they stay within the processing capabilities of the current hardware.
> 2. We apply our VQGLA model to these sample subgraphs, then combine the obtained information of these subgraphs together so that the training process overall learns information of the full graph. In this way, for a graph with $N$ nodes, we can sample $m$ subgraphs for training, with each subgraph using at most $k$ qubits. Typically, $N$ qubits require $O(2^N)$ resource, and this training approach allows us to reduce the resource requirements of VQGLA from $O(2^N)$ to $O(m \cdot 2^k)$, where $m,k<N$.

---

> ### Author Response · Authors · 2024-11-19
> **Response to Reviewer xDyW (Part 2)**
>
> - **The experiment of  S-VQGLA**
>
>     - To validate the effectiveness of S-VQGLA, we conducted an experiment on a commonly used dataset, PROTEINS, which contains proteins classified as enzymes or not. This dataset consists of 1,113 graphs, and each graph has between 4 to **620** nodes. We use ```GraphSAINTEdgeSampler``` provided by Pytroch Geometric (PyG) to sample graphs with more than 14 nodes. Specifically, by ```GraphSAINTEdgeSampler(graph, batch_size=7, num_steps = round(graph.edge_index.shape[-1]/14))```, we can extract many appropriately connected subgraphs with at most 14 nodes. Each subgraph can be processed using our 14-qubit VQGLA with 6 blocks. All subgraphs are input as a batch into the VQGLA, and we combine their results to predict the classification of the full graph. The following table shows the graph classification results for our S-VQGLA and other classical GNN algorithms.
>
> | Methods      | Accuracy (%) |
> | ---------------- | --------------- |
> | GraphSAGE [2]      | 73.0          |
> | GIN  [3]            | 76.2          |
> | GAT-GC [4]        | 76.8          |
> | 2-WL-GNN  [5]      | 76.5          |
> | U2GNN  [6]         | 78.5          |
> | UGT  [7]            | 80.1          |
> | **S-VQGLA  (Ours)**  $\quad$  | **80.4**          |
>
> The compared GNNs contain message-passing GNNs, such as GraphSAGE [2], GIN [3], and GAT-GC [4], and higher-order GNN 2-WL-GNN [5] and methods based on *Graph Transformer*, such as U2GNN [6] and UGT [7]. As we have observed, S-VQGLA not only outperforms message-passing-based GNN methods, but also slightly exceeds the recently proposed Graph Transformer-based methods. ***The graph sampling technique enables our approach to be effectively applied to larger graphs under limited qubit resources, while it also can save qubit resources for future computations.*** We have added the above description and experiment to [our updated PDF](https://openreview.net/pdf?id=IQi8JOqLuv), see Section 6 highlighted in blue in the PDF.
>
> [1] GraphSAINT: Graph sampling based inductive learning method, ICLR 2020.
>
> [2] Inductive representation learning on large graphs. NeurIPS 2017.
>
> [3] How Powerful are Graph Neural Networks? ICLR 2019.
>
> [4] Improving Attention Mechanism in Graph Neural Networks via Cardinality Preservation, IJCAI 2020.
>
> [5] A Novel Higher-order Weisfeiler-Lehman Graph Convolution, ACML, 2020.
>
> [6] Universal Graph Transformer Self-Attention Networks, WWW 2022.
>
> [7] Transitivity-Preserving Graph Representation Learning for Bridging Local Connectivity and Role-Based Similarity, AAAI 2024.
>
> > **W2: The explanation of the quantum operations, especially in the auxiliary and graph message interaction layers, could be clearer. The use of quantum gates and entanglement is discussed at a high level, but more detailed explanations and diagrams would make it easier for readers without extensive quantum computing backgrounds to understand the exact role of each component in the quantum circuit.**
>
> Thanks for your suggestion. Here, we provide a simple and intuitive explanation.
>
> - **For graph message interaction layers**, each edge is mapped into a two-qubit quantum gate acted on qubits representing two nodes. For example, here is a graph $G$ with three nodes $a,b,c$, as shown in [this figure](https://postimg.cc/t7TywZjR). After the first feature encoding layer, the quantum state $|\psi\rangle = |\psi_a\rangle \otimes |\psi_b\rangle \otimes |\psi_c\rangle$. If there is an edge connecting nodes $a$ and $b$, the model will apply a two-qubit quantum gate between qubit $q_a$ and qubit $q_b$. Suppose the used two-qubit gate is $R_{ZZ}(\theta) = \exp(-i\theta Z\otimes Z)$ gate, then it is equivalent to multiplying the quantum state by a matrix $U = R_{ZZ}(\theta) \otimes I$. Thus, the quantum state is changed as $$|\psi^\prime\rangle = U|\psi\rangle = (\exp(-i\theta Z\otimes Z) \otimes I)(|\psi_a\rangle \otimes |\psi_b\rangle \otimes |\psi_c\rangle) =  (R_{ZZ}(\theta) \otimes I)(|\psi_{ab}\rangle \otimes |\psi_c\rangle) = |\psi_{ab}^\prime\rangle \otimes |\psi_c\rangle.$$ In other words, the two-qubit gate alters the quantum states corresponding to nodes $a$ and $b$, thereby achieving the goal of information exchange.
>
> - **For auxiliary layers**, inspired by [8], we use additional ancillary qubits to enhance the expressivity of the model. As shown in [this figure](https://postimg.cc/181c3nHP), an auxiliary qubit is connected to all other nodes through two-qubit gates. In this way, the model can increase the width and number of parameters. Note that the auxiliary layer is optional according to the requirements of different tasks.
>
> We have added more detailed explanations of auxiliary and graph message interaction layers to [our updated PDF](https://openreview.net/pdf?id=IQi8JOqLuv).
>
> [8] Expressivity of quantum neural networks, Physical review research, 2021.

---

> ### Author Response · Authors · 2024-11-19
> **Response to Reviewer xDyW (Part 3)**
>
> > **W3&Q3: Although VQGLA is compared to classical GNNs and a few quantum models, it would be beneficial to include more comparisons with state-of-the-art classical GNN architectures that have shown effectiveness in handling complex graph tasks, such as Graph Transformers or other hybrid models. Could the authors include a comparison with more advanced classical GNN models, such as Graph Transformers, to highlight the specific advantages of the quantum-driven approach?**
>
> In Table 5 of the previous pdf, we compared some advanced GNNs, including Graph Transformer [9]. In addition, in Table 8, we compared some state-of-the-art classical GNNs, including Graphormer [10], k-WL hierarchy-based model δ-k-LGNN [11], subgraph-based model NGNN [12], and substructure-based model GSN [13]. It seems that this part of the results was not written clearly enough in the previous PDF, so we have rewritten and highlighted them in [the updated version of the PDF](https://openreview.net/pdf?id=IQi8JOqLuv). Furthermore, we have also provided more comparative results on methods based on graph transformers (U2GNN [5] and UGT [6]) in the *response to W1&Q1*. In the above three experiments, our VQGLA outperforms the advanced GNNs.
>
> [9] Masked label prediction: Unified message passing model for semi-supervised classification. IJCAI 2021.
>
> [10] Do transformers really perform badly for graph representation? NeurLPS 2021.
>
> [11] Weisfeiler and leman go sparse: Towards scalable higher-order graph embeddings, NeurLPS 2021.
>
> [12] Nested graph neural networks, NeurLPS 2021.
>
> [13] Improving graph neural network expressivity via subgraph isomorphism counting, TPAMI 2022.
>
>
> > **W4&Q2. The implementation of VQGLA on near-term quantum devices is briefly mentioned, but the impact of quantum noise on the model's performance is not explored in depth. Quantum hardware is prone to noise, and the absence of a discussion on error mitigation strategies or the robustness of VQGLA under noisy conditions weakens the paper's practical relevance.  2. How does the presence of quantum noise impact the performance of VQGLA? Are there any error mitigation strategies that could be employed?**
>
>
> - **About the impact of noise:** In Table 7 of previous pdf, we studied the impact of noise on the model’s performance on the accuracy of predicting MILP feasibility from 20 qubits to 50 qubits. Specifically, we use Qiskit’s ```matrix_product_state``` simulator, as well as IBM’s ***noise model*** of the backend ```FakeWashingtonV2```, which is built to mimic the behaviors of IBM Quantum systems. The average T1 time of ```FakeWashingtonV2```  is 97.88098536539536 μs, the average T2 time is 95.21640193925937 μs, and the average readout error is 0.02748267716535434. As shown in Table 7, although noise does have an impact on performance, the effect is within an acceptable range.
>
> | Qubit       | 20 (base) $\quad$| 30 $\quad$  | 40  $\quad$ | 50   |
> |-------------|-----------|-------|-------|-------|
> | Noiseless  $\quad$| 1.0       | 0.967 | 0.908 | 0.862 |
> | Noisy | 0.995     | 0.928 | 0.864 | 0.825 |
>
>
> - **About error mitigation:** Our method is orthogonal to various error mitigation strategies and can leverage these strategies to mitigate the impact of noise. Qiskit integrates interfaces for some error mitigation techniques in the Estimator V2 primitive, but they only support execution on real quantum hardware. Therefore, we deployed our circuits on IBM's real quantum hardware using the IBM Quantum platform and applied different error mitigation strategies to mitigate circuit noise. For more experimental details, please refer to the response to the next question Q4.
>
> > **Q4: Is there any practical guidance on implementing VQGLA on near-term quantum hardware? A discussion on the hardware requirements would be very helpful for future work.**
>
> Thanks for your suggestions. Our VQGLA quantum circuit consists of only simple single-qubit and two-qubit gates, making it easy to deploy on existing NISQ devices. To verify this, we use the Qiskit package and the IBM Quantum platform to directly execute our circuits on real IBM quantum hardware. The circuits are first compiled and optimized, then mapped to the real quantum hardware's topology using the ```generate_preset_pass_manager ```function. Subsequently, the expectation values of the quantum circuits are estimated using the ```Estimator V2``` primitive. The Estimator primitive supports three resilience levels:
>
> * Resilience Level 0: No error mitigation techniques are applied.
> * Resilience Level 1: Applies readout error mitigation and measurement twirling using a model-free technique known as Twirled Readout Error eXtinction (TREX) [14].
> * Resilience Level 2: Includes the error mitigation techniques from Level 1 and further applies gate twirling and the Zero Noise Extrapolation (ZNE) [15] method.

---

> ### Author Response · Authors · 2024-11-19
> **Response to Reviewer xDyW (Part 4)**
>
> We conducted experiments using 10 minutes of free usage on the 127-qubit ```IBM Brisbane``` quantum computer, and evaluated the mean square error (MSE) compared to ground truth on MILP dataset with three different scales, as shown in the table below.
>
> |                      | 18 qubit $\quad$ | 26 qubit $\quad$ | 36 qubit |
> | -------------------- | -------- | -------- | -------- |
> | Noise (ibm_brisbane) $\quad$ | 0.2627   | 0.2850   | 0.3461   |
> | resilience_level = 1 | 0.2107   | 0.2428   | 0.3336   |
> | resilience_level = 2 | 0.2077   | 0.2174   | 0.3176   |
>
> The results show that employing error mitigation strategies can improve performance. Higher resilience levels produce more accurate results, but at the cost of increased processing time, which is a trade-off between cost and accuracy. We have added the above experiments to [our updated PDF](https://openreview.net/pdf?id=IQi8JOqLuv), see Appendix J highlighted in blue.
>
> [14] Model-free readout-error mitigation for quantum expectation values, Physical Review A, 2022.
>
> [15] Error mitigation for short-depth quantum circuits. Physical Review Letters, 2017.
>
>
> ---
> We would sincerely appreciate it if you could reconsider your rating if your concerns have been addressed by our rebuttal, and wish to receive your further feedback soon.  If you have any other concerns, we are more than glad to provide further responses.

---

> ### Author Response · Authors · 2024-11-21
> **To Reviewer xDyW**
>
> Dear reviewer xDyW:
>
> We are deeply grateful for your decision to raise the score for our paper (3->6), and we are truly pleased that our response addressed your concerns and questions. Thank you for the time and effort you dedicated to reviewing our work.
>
> With gratitude,

---

### Author Response · Authors · 2024-12-02
**General Response**

Dear area chair and reviewers,

We would like to express our sincere gratitude to all the reviewers for their valuable feedback, which has helped improve our submission.  We are particularly grateful for the reviewers' recognition of our paper as well-thought-out (xDyW), comprehensive (xDyW, x2kb), compelling (xDyW), and interesting (x2kb). In our response to the reviewers, we have thoroughly addressed questions and concerns. Additionally, we have uploaded the revised manuscript to the OpenReview system to address the reviewers' concerns, with all changes highlighted in blue. Below, we provide a summary of the key revisions:

- **The complexity analysis and scalability experiments (xDyW, x2kb).** In Section 6 of our revised manuscript, we have provided gate and qubit complexity analysis of our method. Since our method only involves simple single-qubit and two-qubit gates, along with lightweight angle encoding and Pauli-Z measurement readout, its overall complexity is lower compared to models based on quantum linear algebra or hybrid quantum-classical layers. Furthermore, to address the limitations of quantum resources in the current NISQ (Noisy Intermediate-Scale Quantum) era, we use a graph sampling strategy to apply our method to large-scale graph instances, and conduct experiments on commonly used graph classification datasets. Among them, the number of nodes is up to 620. As shown in Table 9, the result shows that our VQGLA can still achieve good performance on the larger-scale graph dataset.

- **Comparison with more advanced classical GNN models (xDyW, 9Vep).** In Table 8 and Table 9 of our revised manuscript, we have presented comparisons with more advanced classical GNN models, including subgraph-based GNNs, K-WL GNNs, random GNNs, Transformer-based GNNs, and substructure-based GNNs. The experimental results demonstrate that our method can still be superior compared to higher-order GNNs.

- **Noise mitigation experiment and real device experiment (xDyW).** In Table 7, we studied the impact of noise on the model’s performance. In Table 12 of the revised manuscript, we conducted our quantum model on a 127-qubit IBM *Brisbane* quantum computer using the IBM Quantum cloud platform and used three resilience levels to mitigate noise. The results demonstrate that employing error mitigation strategies can improve the performance of our model.


Best regards,

Author of submission 2442

---

### Meta-Review · Area_Chair_UStG · 2024-12-19

**Metareview:**

This paper introduces a quantum-driven graph learning approach that utilizes quantum machine learning (QML) to identify patterns challenging for classical methods. Experimental results highlight the method's effectiveness in solving mixed-integer linear programs (MILPs). Most of the reviewers agreed that the proposed approach is novel and this paper is technically solid. Most of the reviews tend to accept the paper during the reviewer-author discussion phase. Therefore, I recommend this paper to the ICLR 2025 conference.

**Additional Comments On Reviewer Discussion:**

Reviewers xDyW, x2kb, 9Vep rated this paper as 6: marginally above the acceptance threshold (raised to 6), 8: accept (keep the score), and 5: borderline reject, respectively.

The reviewers raised the following concerns.

- Comparisons with state-of-the-art classical GNN architectures (Reviewers xDyW and 9Vep).
- Clarification on the details of the methods (Reviewers xDyW and x2kb).
- Discussion on the scalability and quantum circuit complexity for general graph structures (Reviewer x2kb).

By additional experiments and more details in the rebuttal, the authors address some concerns about scalability, unclear details, insufficient baselines, and scalability. Therefore, I will recommend accepting this paper in its current state.

---

### Decision · Program_Chairs · 2025-01-22

Accept (Poster)